# INTERPRETING ROBUSTNESS PROOFS OF DEEP NEURAL NETWORKS

**Debangshu Banerjee**[1]**, Avaljot Singh**[1]**, Gagandeep Singh**[1,2]
[1]University of Illinois Urbana-Champaign,   [2]VMware Research
`{db21, avaljot2, ggnds}@illinois.edu`

## ABSTRACT

In recent years numerous methods have been developed to formally verify the robustness of deep neural networks (DNNs). Though the proposed techniques are effective in providing mathematical guarantees about the DNNs' behavior, it is not clear whether the proofs generated by these methods are human-understandable. In this paper, we bridge this gap by developing new concepts, algorithms, and representations to generate human understandable insights into the internal workings of DNN robustness proofs. Leveraging the proposed method, we show that the robustness proofs of standard DNNs rely more on spurious input features as compared to the proofs of DNNs trained to be robust. Robustness proofs of the provably robust DNNs filter out a larger number of spurious input features as compared to adversarially trained DNNs, sometimes even leading to the pruning of semantically meaningful input features. The proofs for the DNNs combining adversarial and provably robust training tend to achieve the middle ground.

## 1 INTRODUCTION

The black box construction and lack of robustness of deep neural networks (DNNs) are major obstacles to their real-world deployment in safety-critical applications like autonomous driving (Bojarski et al., 2016) or medical diagnosis (Amato et al., 2013). To mitigate the lack of trust caused by black-box behaviors, there has been a large amount of work on interpreting individual DNN predictions to gain insights into their internal workings. Orthogonally, the field of DNN verification has emerged to formally prove or disprove the robustness of DNNs in a particular input region capturing an infinite set of inputs. Verification can be leveraged to train more robust models (Balunovic & Vechev, 2020).

We argue that while DNN verification does improve reliability to a certain degree, it does not give any semantically meaningful insights into the working of the generated proof. This is in contrast to standard program verification tasks where proofs capture the semantics of the program and the verified property, like deducing the inductive invariants while proving the termination of a program (Alias et al., 2010). Such deductions can be helpful in order to improve the trust of the system and enhance interpretability. Existing techniques (Bau et al., 2017; Nguyen et al., 2016; Mordvintsev et al., 2015) can extract neuron-level details to generate a better understanding of the DNN only on individual inputs. DNN verification techniques lift the reasoning on individual inputs to an infinite set of inputs, usually represented in terms of constraints. So, instead of generating a single output, these methods generate elaborate complex proofs that certify the DNN's correctness. The primary objective of this work is to dissect and interpret the proof generated by a DNN verifier.

**Key Challenges.** The proofs generated by state-of-the-art DNN verifiers (DeepZ (Singh et al., 2018a), CROWN (Zhang et al., 2018) etc.) encode high-dimensional complex convex shapes defined over thousands of neurons in the DNN, thereby making it difficult to interpret them. Therefore, it is imperative to dissect the proof into smaller pieces called *proof features* that can be interpreted independently and look only at a subset of more important proof features. This subset should be less intricate than the original proof while still effectively approximating it. The key challenge here is to identify different features of the proof, concretize the importance of each of these features, and measure how well a set of proof features can approximate the original proof. As we describe in Section 4, this leads to an intricate tradeoff - size vs. effective approximation vs. the importance of a proof feature set.

**Our Contributions.** We make the following contributions to overcome these challenges:

- We introduce a novel concept of proof features by dissecting a complex high-dimensional proof into smaller parts that can be interpreted independently. We propose a priority function over the proof features that signify the importance of each proof feature w.r.t complete proof.
- For proof interpretation, first we design a general algorithm - ***ProFIt*** (**Pro**o**F** **I**nterpretation **T**echnique) for extracting a set of proof features that is (a) small (easy to interpret), (b) retains only the more important parts of the proof, and (c) preserves the property (Section 4). Finally, we adapt the existing visualization techniques (Sundararajan et al., 2017) to map extracted proof features to human-understandable visualizations (Section 4.4).
- Using the proposed framework, we observe an interesting discrepancy in the semantic meanings captured by the proofs for DNNs trained using different techniques. We compare visualizations of the extracted proof features for standard DNNs and state-of-the-art robustly trained DNNs for the MNIST and CIFAR10 datasets. We observe that the proof features for the standard DNNs rely on spurious regions of the input images while the proof features of adversarially trained DNNs (Madry et al., 2018) filter out some of the spurious parts of the input. Further, the proofs for networks trained with certifiable training (Zhang et al., 2020) do not rely on any spurious parts of the input but they also miss out on some meaningful parts. Proofs for training methods that combine both empirical and certified robustness (Balunovic & Vechev, 2020) provide a common ground. They not only preserve meaningful input parts but also selectively filter out spurious ones. We empirically show that these observations are not contingent on any specific DNN verifier.

We believe that trustworthy DNNs should not only be robust, but their robustness proofs should capture human-aligned internal representations. The feedback from ProFIt can be leveraged as a criterion to differentiate among DNNs achieving similar levels of robustness. ProFIt code is available at https://github.com/uiuc-focal-lab/Profit.

## 2 RELATED WORK

**DNN interpretability and debugging.** There has been an extensive effort to develop interpretability tools for investigating and debugging the internal workings of DNNs. These include feature attribution techniques like saliency maps (Sundararajan et al., 2017; Smilkov et al., 2017), using surrogate models to interpret local decision boundaries (Ribeiro et al., 2016), finding influential (Koh & Liang, 2017), prototypical (Kim et al., 2016), or counterfactual inputs (Goyal et al., 2019), training sparse decision layers (Wong et al., 2021), utilizing robustness analysis (Hsieh et al., 2021). Most of these interpretability tools focus on generating local explanations that investigate how DNNs work on individual inputs. Another line of work, instead of explaining individual inputs, identifies specific concepts associated with a particular neuron (Simonyan et al., 2014; Bau et al., 2020). However, to the best of our knowledge, there is no existing work for interpreting DNN robustness proofs.

**DNN verification.** Unlike DNN interpretability methods, prior works in DNN verification focus on formally proving whether a DNN satisfies desirable properties like robustness (Singh et al., 2019c; Wang et al., 2021b), fairness (Mazzucato & Urban, 2021), etc. The DNN verifiers are broadly categorized into three main categories - (i) sound but incomplete verifiers which may not always prove property even if it holds (Gehr et al., 2018; Singh et al., 2018a; 2019b;a; Zhang et al., 2018; Xu et al., 2020; Salman et al., 2019), (ii) complete verifiers that can always prove the property if it holds (Wang et al., 2018; Gehr et al., 2018; Bunel et al., 2020a;b; Bak et al., 2020; Ehlers, 2017; Ferrari et al., 2022a; Fromherz et al., 2021; Wang et al., 2021a; Palma et al., 2021; Anderson et al., 2020; Zhang et al., 2022) and (iii) verifiers with probabilistic guarantees (Cohen et al., 2019).

**Robustness training.** Existing works (Madry et al., 2018; Balunovic & Vechev, 2020; Zhang et al., 2020) in developing robust training methods for neural networks provide a framework to produce networks that are inherently immune to adversarial perturbations in input.

## 3 PRELIMINARIES

In this section, we provide the necessary background on DNN verification and existing works on debugging DNNs with sparse decision layers. While our method is applicable to general architectures, for simplicity, we focus on a $l$-layer feedforward DNN $N : \mathbb{R}^{d_0} \to \mathbb{R}^{d_l}$ for the rest of this paper. Each layer $i$ except the final one applies the transformation $X_i = \sigma_i(W_i \cdot X_{i-1} + B_i)$ where $W_i \in \mathbb{R}^{d_i \times d_{i-1}}$ and $B_i \in \mathbb{R}^{d_i}$ are respectively the weights and biases of the affine transformation and $\sigma_i$ is a non-linear activation like ReLU, Sigmoid, etc. corresponding to layer $i$. The final layer only applies the affine transformation and the network output is a vector $Y = W_l \cdot X_{l-1} + B_l$.

**DNN verification.** At a high level, DNN verification involves proving that the network outputs $Y = N(X)$ corresponding to all inputs $X$ from an input region specified by $\phi$, satisfy a logical

specification $\psi$. For DNN robustness, the output specification is written as $\psi(Y) = (C^T Y \geq 0)$ where $C \in \mathbb{R}^{d_l}$ defines the linear inequality for encoding the robustness property. For the rest of the paper, we refer to the input region $\phi$ and output specification $\psi$ together as *property* $(\phi, \psi)$. Next, we briefly discuss how DNN verifiers work. A DNN verifier $\mathcal{V}$ computes a possibly over-approximated output region $\mathcal{A} \subseteq \mathbb{R}^{d_l}$ containing all possible outputs of $N$ corresponding to $\phi$. Let, $\Lambda(\mathcal{A}) = \min_{Y \in \mathcal{A}} C^T Y$ denote the minimum value of $C^T Y$ where $Y \in \mathcal{A}$. Then $N$ satisfies property $(\phi, \psi)$ if $\Lambda(\mathcal{A}) \geq 0$. Most existing DNN verifiers (Zhang et al., 2018), for non-linear activation functions, compute convex regions that over approximate the output of the activation function. Note that, due to the over-approximations, DNN verifiers are sound but not complete - the verifier may not always prove property even if it holds. For piecewise linear activation functions like ReLU, complete verifiers exist handling the activation exactly. Nevertheless, complete verification in the worst case takes exponential time. In the rest of the paper, we primarily focus on deterministic, sound, and incomplete verifiers which are more scalable than complete verifiers.

**Analyzing DNN inferences with sparse decision layer.** For analyzing individual DNN inferences, only looking at the output of the DNN hides key details about how the final output was computed by combining the output of each individual neuron (Nguyen et al., 2016; Wong et al., 2021). Conversely, DNNs considered in this paper, have complex multi-layer structures, making them harder to analyze while retaining per neuron-level information. Instead of analyzing neurons from each layer of the DNN, recent works (Wong et al., 2021; Liao & Cheung, 2022) treat DNNs as the composition of a *deep feature extractor* and an affine *decision layer*. The output of each neuron of the penultimate layer represents a single *deep feature* and the final affine layer linearly combines these deep features to compute the network output. This perspective enables us to identify the set of deep features used by the network to compute its output and to investigate their semantic meaning using the existing feature visualization techniques (Ribeiro et al., 2016). However, visualizing each deep feature is practically infeasible for large DNNs where the penultimate layer can contain thousands of neurons. To address this, the work of (Wong et al., 2021) tries to identify a smaller set of features that are sufficient to maintain the performance of the network. This smaller but sufficient feature set retains only the most important features corresponding to a given input. (Wong et al., 2021) empirically shows that a small subset of deep features ($\leq 10$) is sufficient to maintain the accuracy of state-of-the-art models.

## 4 INTERPRETING DNN ROBUSTNESS PROOFS

Next, we describe our approach for interpreting DNN robustness proofs.

**Proof features.** Similar to traditional methods used for interpreting DNN inferences described above (Wong et al., 2021), for interpreting proofs, we propose to segregate the final affine layer from the network and look at the neuron-level features extracted at the penultimate layer. However, proofs generated by DNN verifiers are over an input region ($\phi$) consisting of infinitely many inputs instead of a single input as handled by existing works. In this case, for a given input region $\phi$, we look at the symbolic shape (for example - zonotopes, polytopes, etc.) computed by the verifier at the penultimate layer and then compute its projection on each dimension of the penultimate layer. These projections yield an interval $[l_n, u_n]$ containing all possible output values of the corresponding neuron $n$ w.r.t $\phi$.

**Definition 1** (Proof Features). *Given a network $N$, input region $\phi$ and neural network verifier $\mathcal{V}$, for each neuron $n_i$ at the penultimate layer of $N$, the proof feature $\mathcal{F}_{n_i}$ extracted at that neuron $n_i$ is an interval $[l_{n_i}, u_{n_i}]$ such that $\forall X \in \phi$, the output at $n_i$ always lies in the range $[l_{n_i}, u_{n_i}]$.*

Note that, the computation of the proof features is verifier dependent, i.e., for the same network and input region, different verifiers may compute different values $l_n$ and $u_n$ for a particular neuron $n$. Similarly, even two output-equivalent networks that produce the same output on all possible inputs but have different architectures can have different proofs and subsequently different proof features. For any input region $\phi$, the first $(l-1)$ layers of $N$ along with the verifier $\mathcal{V}$ act as the *proof feature extractor*. For the rest of this paper, we use $\mathcal{F}$ to denote the set of all proof features at the penultimate layer and $\mathcal{F}_I$ to denote the proof features corresponding to $I \subseteq [d_{l-1}]$ i.e. $\mathcal{F}_I = \{\mathcal{F}_{n_i} \mid i \in I\}$. Suppose $N$ is formally verified by the verifier $\mathcal{V}$ to satisfy the property $(\phi, \psi)$. Then in order to gain insights about the proof generated by $\mathcal{V}$, we can directly investigate (described in section 4.4) all the extracted proof features $\mathcal{F}$. However, the number of proof features for contemporary networks can be very large (in thousands). Many of these proof features may not be important for the proof. Similar to how DNNs are interpreted w.r.t individual inferences, we want to identify a smaller set of proof features that are more important for the proof of the property $(\phi, \psi)$.

### 4.1 PROBLEM FORMULATION FOR PROOF INTERPRETATION

Before delving into the details, first, we lay out our expectations for the proof feature set that will be used for interpreting the proof and explain why these expectations are relevant. In section 5.2, we empirically validate that the proof feature set extracted by ProFIt fulfills these expectations.

**A. Small size.** The size of the proof feature set $\mathcal{F}_S$ should be minimized so that investigating the constituent proof features becomes easier. Otherwise, investigating the entire set $\mathcal{F}$ is always a valid but expensive option considering the size of $\mathcal{F}$ is large (in thousands) for contemporary DNNs.

**B. Sufficiency.** Beyond the small size, we argue that any candidate proof feature set $\mathcal{F}_S \subseteq \mathcal{F}$ must at least prove the property $(\phi, \psi)$ with verifier $\mathcal{V}$. Otherwise, it does not make sense to interpret a proof feature set that itself does not satisfy the property. Next, we introduce the novel concepts of proof feature pruning to formally define sufficient proof feature set below:

**Definition 2** (Proof feature Pruning). *Pruning any proof feature $\mathcal{F}_{n_i} \in \mathcal{F}$ corresponding to neuron $n_i$ in the penultimate layer involves setting weights of all its outgoing connections to 0 so that given any input $X \in \phi$ the final output of $N$ no longer depends on the output of $n_i$.*

Once, a proof feature $\mathcal{F}_{n_i}$ is pruned the verifier $\mathcal{V}$ no longer uses $\mathcal{F}_{n_i}$ to prove the property $(\phi, \psi)$.

**Definition 3** (Sufficient proof feature set). *For the proof of property $(\phi, \psi)$ on DNN $N$ with verifier $\mathcal{V}$, a nonempty set $\mathcal{F}_S \subseteq \mathcal{F}$ of proof features is sufficient if the property still holds with verifier $\mathcal{V}$ even all the proof features **not in** $\mathcal{F}_S$ are pruned.*

Before detailing other properties we expect the proof feature set used for interpreting the proof to satisfy, we explain how we can algorithmically check whether any proof feature set $\mathcal{F}_S$ is sufficient or not. Let, $W_l[:, i] \in \mathbb{R}^{d_l}$ denote the $i$-th column of the weight matrix $W_l$ of the final layer $N_l$. Pruning any proof feature $\mathcal{F}_{n_i}$ results in setting all weights in $W_l[:, i]$ to 0. For any proof feature set $\mathcal{F}_S \subseteq \mathcal{F}$, let $W_l(S) \in \mathbb{R}^{d_l \times d_{l-1}}$ be the weight matrix of the pruned final layer that only retains proof features corresponding to $\mathcal{F}_S$. Then columns of $W_l(S)$ are defined below where $\underline{0}$ is a constant all-zero vector

$$W_l(S)[:, i] = \begin{cases} W_l[:, i] & i \in S \\ \underline{0} & \text{otherwise} \end{cases} \tag{1}$$

The proof feature set $\mathcal{F}_S$ is sufficient iff the property $(\phi, \psi)$ can be verified by $\mathcal{V}$ on $N$ with the pruned weight matrix $W_l(S)$. Let, the verifier $\mathcal{V}$ compute an over-approximated output region $\mathcal{A}$ of $N$ over the input region $\phi$. Given that we never change the input region $\phi$ and the proof feature extractor composed of the first $l-1$ layers of $N$ and the verifier $\mathcal{V}$, the output region $\mathcal{A}$ of the pruned network only depends on the pruning done at the final layer. Now let $\mathcal{A}(W_l, S)$ denote the over-approximated output region corresponding to $W_l(S)$. The neural network $N$ can be verified by $\mathcal{V}$ on the property $(\phi, \psi)$ with $W_l(S)$ iff the lower bound $\Lambda(\mathcal{A}(W_l, S)) \geq 0$.

**C. Importance.** The notion of sufficiency and minimality of $\mathcal{F}_S$ alone does not enforce that $\mathcal{F}_S$ always includes proof features that are "important" for the proof. To resolve this issue, we define priority $P(\mathcal{F}_{n_i})$ for every proof feature $\mathcal{F}_{n_i}$ that captures its importance w.r.t the proof. Given a sufficient proof features set $\mathcal{F}_S$, for a proof feature $\mathcal{F}_{n_i} \in \mathcal{F}_S$, we compute its priority by estimating the absolute change $\Delta(\mathcal{F}_{n_i}, \mathcal{F}_S)$ that occurs to $\Lambda(\mathcal{A}(W_l, S))$ if $\mathcal{F}_{n_i}$ is pruned from $\mathcal{F}_S$. At a high level, for each proof feature $\mathcal{F}_{n_i}$ contained in a sufficient feature set, the priority of $\mathcal{F}_{n_i}$ tries to estimate whether pruning $\mathcal{F}_{n_i}$ violates the property $(\phi, \psi)$ or not. Pruning a proof feature $\mathcal{F}_{n_i}$ with a small $\Delta(\mathcal{F}_{n_i}, \mathcal{F}_S)$ only results in a small absolute change in the verifier output and the remaining proof features $\mathcal{F}_S \setminus \{\mathcal{F}_{n_i}\}$ will likely satisfy the property. Moreover, in this case, the set $\mathcal{F}_S \setminus \{\mathcal{F}_{n_i}\}$ precisely approximate (the absolute change is small) the verifier output $\Lambda(\mathcal{A}(W_l, S))$ computed with the entire set $\mathcal{F}_S$. Let, the over-approximated output region computed by verifier $\mathcal{V}$ corresponding to $\mathcal{F}_S \setminus \{\mathcal{F}_{n_i}\}$ be $\mathcal{A}(W_l, S \setminus \{i\})$ then $\Delta(\mathcal{F}_{n_i}, \mathcal{F}_S)$ is defined as follows

$$\Delta(\mathcal{F}_{n_i}, \mathcal{F}_S) = |\Lambda(\mathcal{A}(W_l, S)) - \Lambda(\mathcal{A}(W_l, S \setminus \{i\}))|$$

However, $\Delta(\mathcal{F}_{n_i}, \mathcal{F}_S)$ depends on the particular sufficient proof set $\mathcal{F}_S$ and does not estimate the global importance of $\mathcal{F}_{n_i}$ independent of the choice of $\mathcal{F}_S$. Hence to define the priority $P(\mathcal{F}_{n_i})$ of a proof feature $\mathcal{F}_{n_i}$ we take the maximum of $\Delta(\mathcal{F}_{n_i}, \mathcal{F}_S)$ over all sufficient feature sets $\mathcal{F}_S$ containing $\mathcal{F}_{n_i}$. Let, $\mathbb{S}(\mathcal{F}_{n_i})$ denote set of all sufficient $\mathcal{F}_S$ containing $\mathcal{F}_{n_i}$. Then, $P(\mathcal{F}_{n_i})$ is as follows

$$P(\mathcal{F}_{n_i}) = \max_{\mathcal{F}_S \in \mathbb{S}(\mathcal{F}_{n_i})} \Delta(\mathcal{F}_{n_i}, \mathcal{F}_S) \tag{2}$$

Overall, for proof interpretation, we want to extract a proof feature set $\mathcal{F}_{S_0}$ that is as small as possible, sufficient, and retains important proof features. However, there is a tradeoff between the three criteria.

For example, the entire proof feature set $\mathcal{F}$ is always sufficient but it is not small. Similarly, if $C^T B_l \geq 0$ ($B_l$ is the bias of the final layer) the minimum sized sufficient proof feature set will always be empty. In this case, although the empty set is the smallest and sufficient it does not retain any important proof features. This tradeoff between size, sufficiency, and importance makes it hard to find a $\mathcal{F}_{S_0}$. Moreover, the search space for $\mathcal{F}_{S_0}$ is prohibitively large containing $2^{d_{l-1}}$ possible candidates, and computing $\mathcal{F}_{S_0}$ with an exhaustive search is infeasible. Even just computing the priority $P(\mathcal{F}_{n_i})$ for a proof feature $\mathcal{F}_{n_i}$ can be expensive considering the set $\mathbb{S}(\mathcal{F}_{n_i})$ can be exponentially large. So, we first design an approximation of $P(\mathcal{F}_{n_i})$ that is easy to compute.

### 4.2 APPROXIMATE PRIORITY OF PROOF FEATURES

As described above, finding the maximum value of $\Delta(\mathcal{F}_{n_i}, \mathcal{F}_S)$ over $\mathbb{S}(\mathcal{F}_{n_i})$ is practically infeasible. So we compute an upper bound $P_{ub}(\mathcal{F}_{n_i})$ of $P(\mathcal{F}_{n_i})$ by estimating a global non-trivial upper bound of $\Delta(\mathcal{F}_{n_i}, \mathcal{F}_S)$, that holds $\forall \mathcal{F}_S \in \mathbb{S}(\mathcal{F}_{n_i})$. The proposed global upper bound is independent of the choice of $\mathcal{F}_S \in \mathbb{S}(\mathcal{F}_{n_i})$ and therefore removes the need to iterate over $\mathbb{S}(\mathcal{F}_{n_i})$ enabling efficient computation. For the network, $N$ and input region $\phi$, let $\mathcal{A}_{l-1}$ denote the over-approximate symbolic region computed by $\mathcal{V}$ at the penultimate layer. Then $\forall \mathcal{F}_S \in \mathbb{S}(\mathcal{F}_{n_i})$ the global uppper bound of $\Delta(\mathcal{F}_{n_i}, \mathcal{F}_S)$ can be computed as follows where for any $X \in \mathbb{R}^{d_{l-1}}$, $x_i$ denotes its $i$-th coordinate:

$$\Delta(\mathcal{F}_{n_i}, \mathcal{F}_S) \leq \max_{X \in \mathcal{A}_{l-1}} |(C^T W_l(S) X - C^T W_l(S \setminus \{i\}) X)| = \max_{X \in \mathcal{A}_{l-1}} |(C^T W_l[:, i]) \cdot x_i|$$

$$P(\mathcal{F}_{n_i}) = \max_{\mathcal{F}_S \in \mathbb{S}(\mathcal{F}_{n_i})} \Delta(\mathcal{F}_{n_i}, \mathcal{F}_S) \leq \max_{X \in \mathcal{A}_{l-1}} |(C^T W_l[:, i]) \cdot x_i|$$

Now, any proof feature $\mathcal{F}_{n_i} = [l_{n_i}, u_{n_i}]$ computed by $\mathcal{V}$ contains all possible values of $x_i$ where $X \in \mathcal{A}_{l-1}$. Leveraging this observation, we can further simplify the upper bound $P_{ub}(\mathcal{F}_{n_i})$ of $P(\mathcal{F}_{n_i})$ as shown below.

$$P(\mathcal{F}_{n_i}) \leq \max_{x_i \in [l_{n_i}, u_{n_i}]} |(C^T W_l[:, i])| \cdot |x_i| = |(C^T W_l[:, i])| \cdot \max(|l_{n_i}|, |u_{n_i}|) = P_{ub}(\mathcal{F}_{n_i}) \quad (3)$$

This simplification ensures that $P_{ub}(\mathcal{F}_{n_i})$ for all $\mathcal{F}_{n_i}$ can be computed with $O(d_{l-1})$ elementary vector operations and a single verifier call that computes the intervals $[l_{n_i}, u_{n_i}]$.

### 4.3 PROFIT ALGORITHM

Next, we describe how we approximately compute $\mathcal{F}_{S_0}$ that is sufficient, small in size, and retains proof features $\mathcal{F}_{n_i}$ with higher priorities $P_{ub}(\mathcal{F}_{n_i})$. To reduce the size of the proof feature set, a trivial step is to just prune all the proof features from $\mathcal{F}$ whose $P_{ub}$ is 0. These features do not have any contribution to the proof of the property $(\phi, \psi)$ by the verifier $\mathcal{V}$. This step forms a trivial algorithm. However, this is not enough. We can further prune more proof features leading to a yet smaller set. For this, we propose an iterative algorithm **ProFIt** shown in Algorithm 1 which maintains two set namely, $\boldsymbol{F}_{S_0}$ and $\boldsymbol{F}_S$. $\boldsymbol{F}_{S_0}$ contains the features guaranteed to be included in the final answer computed by ProFIt and $\boldsymbol{F}_S$ contains the candidate features yet to be pruned by the algorithm. At every step, the algorithm ensures that the set $\boldsymbol{F}_S \cup \boldsymbol{F}_{S_0}$ is sufficient and iteratively reduces its size by pruning proof features from $\boldsymbol{F}_S$. The algorithm iteratively prunes the feature $\mathcal{F}_{n_i}$ with the lowest value of $P_{ub}(\mathcal{F}_{n_i})$ from $\boldsymbol{F}_S$ while retaining features with higher priorities in $\boldsymbol{F}_S \cup \boldsymbol{F}_{S_0}$. At Line 8 in the algorithm, $\boldsymbol{F}_{S_0}$ and $\boldsymbol{F}_S$ are initialized as empty set ({}) and $\mathcal{F}$ respectively. We note that checking the sufficiency of any arbitrary proof feature set $\mathcal{F}_S$ (Definition 3) is not trivial and requires expensive verifier invocations. Since we are modifying only the final layer, we use incremental verification (Ugare et al., 2022; 2023; 2024) that can efficiently verify the property without starting from scratch. Still removing a single feature in each iteration and checking the sufficiency of the remaining features in the worst case leads to $O(d_{l-1})$ incremental verification calls which are expensive. Instead, at each step, from $\boldsymbol{F}_S$ our algorithm greedily picks top-$|S|/2$ features (line 10) $\boldsymbol{F}_{S_1}$ based on their priority and invokes the verifier $\mathcal{V}$ to check the sufficiency of $\boldsymbol{F}_{S_0} \cup \boldsymbol{F}_{S_1}$ (line 12). If the feature set $\boldsymbol{F}_{S_0} \cup \boldsymbol{F}_{S_1}$ is sufficient (line 13), ProFIt removes all features in $\boldsymbol{F}_S \setminus \boldsymbol{F}_{S_1}$ from $\boldsymbol{F}_S$ and therefore $\boldsymbol{F}_S$ is updated as $\boldsymbol{F}_{S_1}$ in this step (line 14). Otherwise, if $\boldsymbol{F}_{S_0} \cup \boldsymbol{F}_{S_1}$ does not preserve the property $(\phi, \psi)$ (line 15), ProFIt adds all feature in $\boldsymbol{F}_{S_1}$ to $\boldsymbol{F}_{S_0}$ (line 16) and replaces $\boldsymbol{F}_S$ with $\boldsymbol{F}_S \setminus \boldsymbol{F}_{S_1}$ (line 17). The algorithm terminates after all features in $\boldsymbol{F}_S$ are exhausted. Since at every step, the algorithm reduces size of $\boldsymbol{F}_S$ by half, it always terminates within $O(\log(d_{l-1}))$ incremental verifier calls. As mentioned before, the empty proof feature set is sufficient iff $C^T B_l \geq 0$, in this case, ProFIt extracts smallest *non-empty* sufficient proof feature set which like all other cases includes proof features with the higher priority. In Appendix A, we derive mathematical guarantees about the correctness and efficacy

---

**Algorithm 1** Approximate minimum proof feature extraction

1: **Input:** DNN $N$, property $(\phi, \psi)$, verifier $\mathcal{V}$.
2: **Output:** approx. minimum proof features $\boldsymbol{F}_{S_0}$,
3: **if** $\mathcal{V}$ can not verify $N$ on $(\phi, \psi)$ **then**
4:    **return**
5: **end if**
6: Calculate all proof features for input region $\phi$.
7: Calculate priority $P_{ub}(\mathcal{F}_{n_i})$ all proof features.
8: **Initialization:** $\boldsymbol{F}_{S_0} = \{\}$, $\boldsymbol{F}_S = \mathcal{F}$
9: **while** $\boldsymbol{F}_S$ is not empty **do**
10:    $\boldsymbol{F}_{S_1} = \text{top-}|S|/2$ features based on $P_{ub}(\mathcal{F}_{n_i})$
11:    $\boldsymbol{F}_{S_2} = \boldsymbol{F}_S \setminus \boldsymbol{F}_{S_1}$
12:    Check sufficiency of $\boldsymbol{F}_{S_0} \cup \boldsymbol{F}_{S_1}$ with $\mathcal{V}$
13:    **if** $\boldsymbol{F}_{S_0} \cup \boldsymbol{F}_{S_1}$ is sufficient **then**
14:       $\boldsymbol{F}_S = \boldsymbol{F}_{S_1}$
15:    **else**
16:       $\boldsymbol{F}_{S_0} = \boldsymbol{F}_{S_0} \cup \boldsymbol{F}_{S_1}$
17:       $\boldsymbol{F}_S = \boldsymbol{F}_{S_2}$
18:    **end if**
19: **end while**
20: **return** proof features $\boldsymbol{F}_{S_0}$.

---

of Algorithm 1. For correctness, we prove that the feature set $\boldsymbol{F}_{S_0}$ is always sufficient (Definition 3). For efficacy, we theoretically find a non-trivial upper bound on the size of $\boldsymbol{F}_{S_0}$.

### 4.4 VISUALIZATION OF EXTRACTED PROOF FEATURES

Once the proof features are extracted, we want to map them to a human-interpretable format. There exists a plethora of works (Sundararajan et al., 2017; Smilkov et al., 2017) that generate human-interpretable visualizations of the output of individual neurons w.r.t single inputs. However, these techniques are insufficient to generate explanations w.r.t an input region. To resolve this, we adapt the existing local gradient-based visualization techniques (Sundararajan et al., 2017) for visualizing the extracted proof features. As shown in existing works (Mirman et al., 2018), common incomplete DNN verifiers (Xu et al., 2021; Singh et al., 2019b) can be represented as differentiable programs. This enables us to use the gradients on the differentiable programs for visualization. For any proof feature $\mathcal{F}_n = [l_n, u_n]$ both $l_n, u_n$ can be expressed as differentiable functions $l_n = L_n(x_l^1, x_u^1, \ldots, x_l^{d_0}, x_u^{d_0})$ and $u_n = U_n(x_l^1, x_u^1, \ldots, x_l^{d_0}, x_u^{d_0})$ where $\forall i \in d_0$ $x_l^i = x_i - \epsilon_i$ and $x_u^i = x_i + \epsilon_i$ are the lower and upper bound of the $i$-th input cooridinate, $x_i$ is the unperturbed value, $\epsilon_i$ is the amount of perturbation. To measure the sensitivity of proof feature $\mathcal{F}_n$ w.r.t change in $i$-th input coordinate, we take the gradient $\frac{1}{2} \times (\frac{\partial L_n}{\partial \epsilon_i} + \frac{\partial U_n}{\partial \epsilon_i})$ of the mean (also the midpoint) $\frac{(l_n+u_n)}{2}$ of $\mathcal{F}_n$ w.r.t $\epsilon_i$. This gradient captures the change in the mean value of the proof feature w.r.t the change in $i$-th input coordinate. However, complete DNN verifiers such as SMT-based (Katz et al., 2017) or LP/MILP-based verifiers (Singh et al., 2018b) cannot be represented as differentiable programs. For a non-differentiable DNN verifier, we propose an alternative visualization technique in Appendix B based on statistical estimation of the mean gradient over inputs satisfying $\phi$.

### 4.5 LIMITATIONS

The scalability of our method is ultimately limited by the scalability of the existing DNN verifiers. Therefore, ProFIt currently cannot handle large DNNs (e.g. vision transformers (Dosovitskiy et al., 2021), ImageNet classifiers, etc.). Nonetheless, ProFIt is general and compatible with any verification algorithm. Therefore, ProFIt will benefit from any future advances to enable the DNN verifiers to scale to larger DNNs and datasets. We focus on the proof features extracted at the penultimate layer but not at other hidden layers. ProFIt can only handle deterministic verifiers and does not currently work with probabilistic verifiers like - randomized smoothing (Cohen et al., 2019). These extensions will require defining priority for proof features and devising algorithms to extract them.

## 5 EXPERIMENTAL EVALUATION

### 5.1 EXPERIMENTAL SETUP

For evaluation we use convolutional networks trained on two popular datasets - MNIST (LeCun et al., 1989) CIFAR-10 (Krizhevsky, 2009) shown in Table 1. The networks are trained with standard training and three state-of-the-art robust training methodologies - adversarial training (PGD training) (Madry et al., 2018), certified robust training (CROWN-IBP) (Zhang et al., 2020) and a combination of both adversarial and certified training (COLT) (Balunovic & Vechev, 2020). For experiments, we use pre-trained publically available networks - the standard and PGD networks are from the

| Dataset | Training Method | Input Region ($\phi$) eps ($\epsilon$) | No. of proved properties | Original Feature Count | Mean Proof Feature Count | | | No. of proofs with $\leq 5$ proof features (ProFIt) | No. of proofs with $\leq 10$ proof features (ProFIt) |
|---|---|---|---|---|---|---|---|---|---|
| | | | | | Random | Gradient | ProFIt | | |
| **MNIST** | Standard | 0.02 | 459 | 100 | 20.31 | 5.25 | 1.96 | 449 | 457 |
| | PGD Trained | 0.02 | 415 | 1000 | 93.29 | 13.73 | 6.02 | 315 | 364 |
| | COLT | 0.02 | 480 | 100 | 14.45 | 5.43 | 3.46 | 401 | 461 |
| | CROWN-IBP | 0.02 | 482 | 100 | 9.51 | 6.73 | 6.16 | 240 | 401 |
| **MNIST** | PGD Trained | 0.1 | 191 | 1000 | 162.39 | 35.79 | 3.29 | 131 | 149 |
| | COLT | 0.1 | 281 | 100 | 29.57 | 12.22 | 3.16 | 240 | 271 |
| | CROWN-IBP | 0.1 | 473 | 100 | 10.09 | 7.36 | 6.23 | 232 | 384 |
| **CIFAR-10** | Standard | 0.2/255 | 277 | 100 | 30.36 | 18.28 | 11.12 | 127 | 173 |
| | PGD Trained | 0.2/255 | 298 | 100 | 31.22 | 16.58 | 9.74 | 173 | 210 |
| | COLT | 0.2/255 | 267 | 250 | 30.10 | 18.13 | 9.03 | 170 | 204 |
| | CROWN-IBP | 0.2/255 | 265 | 256 | 7.96 | 7.49 | 5.30 | 172 | 221 |
| **CIFAR-10** | PGD Trained | 2/255 | 173 | 100 | 39.57 | 24.46 | 6.19 | 122 | 144 |
| | COLT | 2/255 | 229 | 250 | 34.64 | 23.25 | 7.76 | 146 | 181 |
| | CROWN-IBP | 2/255 | 206 | 256 | 9.41 | 9.21 | 5.10 | 140 | 176 |

Table 1: ProFIt Efficacy Analysis

ERAN project (Singh et al., 2019c), COLT, and CROWN-IBP networks taken from their official repositories (Balunovic & Vechev, 2020; Zhang et al., 2020). Similar to the existing works on DNN verification (Zhang et al., 2018), we use $L_\infty$-based local robustness properties (Carlini & Wagner, 2017). Here, the input region $\phi$ contains all images obtained by perturbing the intensity of each pixel in the input image independently within a bound $\epsilon$. $\psi$ specifies a region where the network's output for the correct class is higher than all other classes. $\epsilon_{train} = 0.1$ and $\epsilon_{train} = 8/255$ are used while training all robustly trained MNIST and CIFAR-10 networks respectively. Unless specified otherwise, the proofs are generated by running the state-of-the-art incomplete verifier $\alpha$-Crown (Xu et al., 2021). We run all experiments on a 16-core 12th-gen i7 machine with 16 GB of RAM.

## 5.2 EFFICACY OF PROFIT ALGORITHM

In this section, we experimentally evaluate the efficacy of ProFIt in terms of the size of the extracted sufficient proof feature sets and also assess the usefulness of the proposed priority ordering (Eq. 3) in extracting important proof features. The size of the extracted sufficient proof feature set captures the ease of interpreting the extracted feature set that preserves the proof. Considering there is no existing work for property-preserving pruning of proof features, we run ProFIt with two natural heuristics that define the priority of the proof features and use them as baselines. These are (i) random ordering of the proof features and (ii) computing the gradient of the verifier output w.r.t each proof feature and sorting the proof features in the decreasing order of the corresponding gradient magnitude. The gradient magnitude-based priority order has already been successfully applied (Lis et al., 2019) to traditional DNN pruning used for reducing the size of the networks. We note that incomplete verifiers like $\alpha$-Crown are differentiable. Moreover, pruning proof features with gradient-based priority heuristics can be viewed as applying existing feature extraction techniques assuming the underlying verifier is differentiable. For each network, we use the first 500 images from their corresponding test sets for defining input specification $\phi$. We conduct experiments with two different $\epsilon$ values to define $L_\infty$ input region. The $\epsilon$ values used for MNIST networks are 0.02 and 0.1 and that for CIFAR-10 networks are 0.2/255 and 2/255. For experiments with high $\epsilon$ values (0.1 for MNIST and 2/255 for CIFAR-10), we omit standard networks as they do not satisfy local robustness properties defined with high $\epsilon$ values. In comparing proof feature sizes, we exclude "zero" features where both lower and upper bounds ($l$ and $u$) are zero, as they can be removed without affecting the verifier's output.

**Mean extracted proof feature set size comparison:** In Table 1, we summarize the results of the experiments for different networks and different $\epsilon$ values. We show that the mean size of the proof feature set computed using ProFIt (column 8) is significantly smaller than that of both the baselines (columns 6 and 7 respectively). This shows proof feature sets computed using ProFIt are easy to interpret. The distributions of extracted proof feature sets size for all DNNs are in Appendix C.1.

**Evaluation of proof feature priority:** Next, we evaluate the efficacy of the priority ordering of proof features (defined in Eq. 3) in identifying important proof features and compare against the random and gradient-based priority ordering defined above. For these experiments, we fix the proof feature set size and compare the extracted proof feature sets with the fixed size based on two metrics - (i) % cases the extracted proof feature set preserves the property $(\phi, \psi)$ that was satisfied initially and

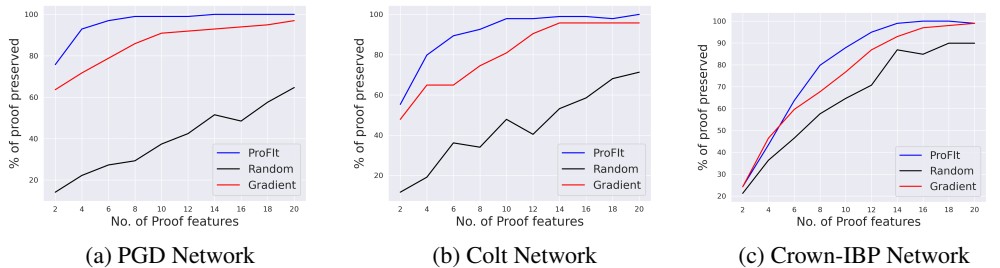

(a) PGD Network        (b) Colt Network        (c) Crown-IBP Network

Figure 1: Percentages of proofs preserved by different priority heuristics on robust MNIST networks.

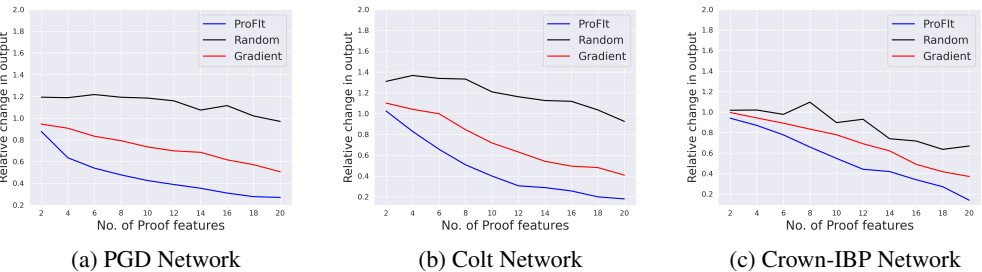

(a) PGD Network        (b) Colt Network        (c) Crown-IBP Network

Figure 2: Relative change in verifier output with different heuristics on robust MNIST networks.

(ii) mean relative change in verifier output after every proof feature, not part of the extracted set is removed. Since we fixed the extracted proof feature set size, the feature set is no longer guaranteed to be sufficient. For proof feature set size varying from 2 to 20, compared to both the baselines, we show that the priority ordering used by ProFIt preserves a higher % of proofs (Fig. 1) while more precisely approximating the original verifier output (the relative change from original verifier output is smaller) (Fig. 2). These plots are generated on robust MNIST networks for $\epsilon = 0.1$. Plots for standard MNIST network and CIFAR-10 networks are shown in Appendix C.2. Additionally, in Appendix C.3 we provide a qualitative evaluation of the priority ordering of proof features where we show that visualizations of proof features with higher priority capture meaningful input features while visualizations of proof features with lower priority are less informative. This shows the proposed priority of the proof features (Eq. 3) is indeed effective in identifying their importance w.r.t the proof.

### 5.3 QUALITITIVE COMPARISON OF ROBUSTNESS PROOFS

(Tsipras et al., 2019) observed that the standardly trained networks rely on some spurious input features to gain a higher accuracy and as a result, are not very robust against adversarial attacks. In contrast, the PGD trained networks rely more on human-understandable features and are, therefore, more robust against attacks. This empirical robustness comes at cost of reduced accuracy. So, there is an inherent dissimilarity between the types of input features that the standard and adversarially trained networks rely on while classifying a single input. Also, certified robust trained networks are even more robust than the empirically trained ones, however, they report even less accuracy (Müller et al., 2021). In this section, we interpret proof features extracted with ProFIt and use the obtained visualizations to qualitatively check for the existence of such dissimilarities among different proofs of the same robustness property on standard and robustly trained networks. We also study the effect of certified robust training methods like CROWN-IBP, empirically robust training methods like PGD, and training methods that combine both adversarial and certified training like COLT on the proof features. For a local input region $\phi$, we say that a robustness proof is semantically meaningful if it focuses on the input features relevant to the output class for images in $\phi$ and not on the spurious features. In the case of MNIST or CIFAR-10 images, spurious input features are the pixels coming from the image background, whereas important input features are the pixels that are a part of the actual object being identified by the network. The gradient map of the extracted proof features w.r.t. the change in input pixels (see Section 4.4) identifies the input pixels that the proof focuses on. In Fig. 3, we compare the gradient maps corresponding to the top proof feature (the one having the highest priority $P_{ub}(\mathcal{F}_{n_i})$) on networks from Table 1 on representative images of different output classes in the MNIST and CIFAR10 test sets. These experiments lead us to interesting observations - even if some property is verified for both the standard network and the robustly trained network, there

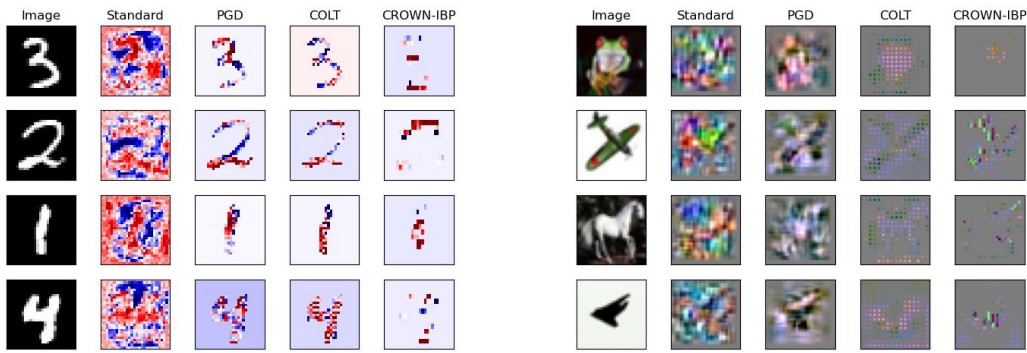

(a) Gradient maps generated on MNIST DNNs.    (b) Gradient maps generated on CIFAR-10 DNNs.

Figure 3: Gradient map corresponding to the top proof feature corresponding to DNNs trained using different methods rely on different input features.

is a difference in the human interpretability of the types of input features that the proofs rely on. The standard networks and the provably robust trained networks like CROWN-IBP are the two extremes of the spectrum. For the standard networks, we observe that although the top-proof feature does depend on some of the semantically meaningful regions of the input image, the gradient at several spurious input features is also non-zero. In contrast, the top proof feature corresponding to state-of-the-art provably robust training method CROWN-IBP filters out most of the spurious features, but it also misses out on some meaningful features. The proofs of PGD-trained networks filter out the spurious features and are, therefore, more semantically aligned than the standard networks. The proofs of the training methods that combine both empirical robustness and provable robustness like COLT in a way provide the best of both worlds by not only selectively filtering out the spurious features but also highlighting the more human interpretable features. So, as the training methods tend to regularize more for robustness, their proofs become more selective in the type of input features that they rely on. To substantiate our observation, we show additional visualizations of the top proof features in Appendix C.4 and visualizations for the entire proof feature set in Appendix C.5. The extracted proof features set and their gradient maps computed w.r.t high $\epsilon$ values ($\epsilon = 0.1$ for MNIST and $\epsilon = 2/255$ for CIFAR-10) are similar to those generated with smaller $\epsilon$ as shown in Appendix C.6.

### 5.4 ABLATION STUDIES

**Different verifiers:** The proof features extracted by ProFIt are specific to the proof generated by the verifier. To show the qualitative analysis presented in section 5.3 is not contingent on $\alpha$-Crown we run ProFIt with different verifiers including DeepZ (Singh et al., 2018a), Crown (Zhang et al., 2018) and state-of-the-art complete verifier $\alpha, \beta$-Crown (Wang et al., 2021b) and compare the top proof feature. We use networks shown in Table 1 and the same properties as before. We observe that in more than 95% of cases, the top proof feature extracted with different verifiers remains the same. Further, the visualizations of the top proof feature w.r.t different differentiable verifiers also align with the observation in section 5.3. In this case, for different verifiers though the proof features $\mathcal{F}_{n_i} = [l_{n_i}, u_{n_i}]$ change, their relative priority ordering computed by Eq. 3 remains the same. Note, while comparing proof features from two different verifiers we only consider those properties that can be proved by both the verifiers. The results are in Appendix C.7 and visualizations in Appendix C.8.

**Training parameters:** In Appendix C.9, we compare proofs generated on networks with the same architecture trained with different training methods to validate that the underlying network architecture does not affect in the conclusions presented in Section 5.3. We also analyze the sensitivity of the extracted proof features to the training parameter $\epsilon_{train}$ that is used to define the $L_\infty$ region during training (Appendix C.10). We observe that networks trained with higher $\epsilon_{train}$ are more robust and the top-proof feature filters out more input features that align with the observations in Section 5.3.

## 6 CONCLUSION

We developed a novel method called ProFIt to interpret DNN robustness proofs. We empirically establish that even if a property holds for a DNN, the proof for the property may rely on spurious or semantically meaningful internal representations depending on the training method. We believe that ProFIt can be applied for diagnosing the trustworthiness of DNNs inside their development pipeline.

## 7 ACKNOWLEDGEMENT

We thank the anonymous reviewers for their insightful comments. This work was supported in part by NSF Grants No. CCF-2238079, CCF-2316233, CNS-2148583, and Google Research Scholar award.

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

# A   THEORETICAL GUARANTEES OF PROFIT

## A.1   PROOF OF SUFFICIENCY OF THE EXTRACTED PROOF FEATURE SET

**Theorem 1.** *If the verifier $\mathcal{V}$ can prove the property $(\phi, \psi)$ on the network $N$, then $\boldsymbol{F}_{S_0}$ computed by Algorithm 1 is sufficient (Definition 3).*

*Proof.* Proof of theorem 1 by induction on the number of steps of the while loop.
**Induction Hypothesis:** At each step of the loop, $\boldsymbol{F}_{S_0} \cup \boldsymbol{F}_S$ is sufficient.
**Base Case:** At step 0, i.e., at initialization, $\boldsymbol{F}_{S_0} = \{\}$ and $\boldsymbol{F}_S = \mathcal{F}$. So, $\boldsymbol{F}_{S_0} \cup \boldsymbol{F}_S = \mathcal{F}$. Given that $\mathcal{V}$ proves the property $(\phi, \psi)$ on $N$, from Definition 3, $\mathcal{F}$ is sufficient.
**Induction Case:** Let $\boldsymbol{F}_{S_0} \cup \boldsymbol{F}_S$ be sufficient for $n$-th step of the loop. Consider the following cases for $(n+1)$-th step of the loop.

1. Let $\boldsymbol{F}_{S_0} \cup \boldsymbol{F}_{S_1}$ be sufficient at line 12. In this case, $\boldsymbol{F}_S$ is updated by $\boldsymbol{F}_{S_1}$ (line 14). So, $\boldsymbol{F}_{S_0} \cup \boldsymbol{F}_S$ is sufficient.

2. Let $\boldsymbol{F}_{S_0} \cup \boldsymbol{F}_{S_1}$ be not sufficient at line 12. In this case, $\boldsymbol{F}_{S_0}$ and $\boldsymbol{F}_S$ are updated as in lines 16 and 17. Let the new $\boldsymbol{F}_{S_0}$ and $\boldsymbol{F}_S$ be $\boldsymbol{F}'_{S_0}$ and $\boldsymbol{F}'_S$. So, $\boldsymbol{F}'_{S_0} = \boldsymbol{F}_{S_0} \cup \boldsymbol{F}_{S_1}$ and $\boldsymbol{F}'_S = \boldsymbol{F}_{S_2}$. So, $\boldsymbol{F}'_{S_0} \cup \boldsymbol{F}'_S = \boldsymbol{F}_{S_0} \cup \boldsymbol{F}_{S_1} \cup \boldsymbol{F}_{S_2}$. Also, $\boldsymbol{F}_{S_1} \cup \boldsymbol{F}_{S_2} = \boldsymbol{F}_S$. So, $\boldsymbol{F}'_{S_0} \cup \boldsymbol{F}'_S = \boldsymbol{F}_{S_0} \cup \boldsymbol{F}_S$. So, from induction hypothesis, $\boldsymbol{F}'_{S_0} \cup \boldsymbol{F}'_S$ is sufficient.

$\square$

## A.2   UPPER BOUND ON THE SIZE OF THE EXTRACTED PROOF FEATURE SET

**Lemma 1.** $\forall S \subseteq [d_{l-1}]$ *if* $i \in S$ *then* $|\Lambda(\mathcal{A}(W_l, S)) - \Lambda(\mathcal{A}(W_l, S \setminus \{i\}))| \leq \max_{X \in \mathcal{A}_{l-1}} |(C^T W_l[:,i]) \cdot x_i|$. *For any vector* $X \in \mathbb{R}^{d_{l-1}}$, $x_i \in \mathbb{R}$ *denotes its $i$-th coordinate.*

*Proof.* Without loss of generality let assume, $\Lambda(\mathcal{A}(W_l, S)) \leq \Lambda(\mathcal{A}(W_l, S \setminus \{i\}))$. Suppose, $\Lambda(\mathcal{A}(W_l, S) = C^T W_l(S) X_{min}$ where $X_{min} \in \mathcal{A}_{l-1}$.

$$\Lambda(\mathcal{A}(W_l, S)) \leq \Lambda(\mathcal{A}(W_l, S \setminus \{i\}))$$
$$C^T W_l(S) X_{min} \leq \Lambda(\mathcal{A}(W_l, S \setminus \{i\})) \leq C^T W_l(S \setminus \{i\}) X_{min}$$
$$|\Lambda(\mathcal{A}(W_l, S)) - \Lambda(\mathcal{A}(W_l, S \setminus \{i\}))| = |C^T W_l(S) X_{min} - \Lambda(\mathcal{A}(W_l, S \setminus \{i\}))|$$
$$\leq |C^T W_l(S) X_{min} - C^T W_l(S \setminus \{i\}) X_{min}|$$
$$\leq \max_{X \in \mathcal{A}_{l-1}} |(C^T W_l[:,i]) \cdot x_i|$$

$\square$

**Lemma 2.** $\forall \mathcal{F}_S \subseteq \mathcal{F}$, $\Delta(\mathcal{F}_S) \leq \sum_{\mathcal{F}_{n_i} \in \mathcal{F} \setminus \mathcal{F}_S} P_{ub}(\mathcal{F}_{n_i})$ *where* $P_{ub}(\mathcal{F}_{n_i})$ *is defined in equation 3 and* $\Delta(\mathcal{F}_S) = |\Lambda(\mathcal{A}) - \Lambda(\mathcal{A}(W_l, S))|$. *For any vector* $X \in \mathbb{R}^{d_{l-1}}$, $x_i \in \mathbb{R}$ *denotes its $i$-th coordinate.*

*Proof.*

$$\Delta(\mathcal{F}_S) = |\Lambda(\mathcal{A}) - \Lambda(\mathcal{A}(W_l, S))|$$

$$\leq \max_{X \in \mathcal{A}_{l-1}} | \sum_{\mathcal{F}_{n_i} \in \mathcal{F} \setminus \mathcal{F}_S} (C^T W[: i]) \cdot x_i|$$

$$\leq \max_{X \in \mathcal{A}_{l-1}} \sum_{\mathcal{F}_{n_i} \in \mathcal{F} \setminus \mathcal{F}_S} |(C^T W[: i]) \cdot x_i|$$

$$\leq \sum_{\mathcal{F}_{n_i} \in \mathcal{F} \setminus \mathcal{F}_S} \max_{X \in \mathcal{A}_{l-1}} |(C^T W[: i]) \cdot x_i|$$

$$= \sum_{\mathcal{F}_{n_i} \in \mathcal{F} \setminus \mathcal{F}_S} P_{ub}(\mathcal{F}_{n_i}) \quad [\text{ From } equation \text{ } 3]$$

$\square$

**Lemma 3.** *A feature set $\mathcal{F}_S \subseteq \mathcal{F}$ with $\Delta(\mathcal{F}_S) \leq \Lambda(\mathcal{A})$ is sufficient provided $\Lambda(\mathcal{A}) \geq 0$ where $\Delta(\mathcal{F}_S) = |\Lambda(\mathcal{A}) - \Lambda(\mathcal{A}(W_l, S))|$.*

*Proof.* $\Delta(\mathcal{F}_S) = |\Lambda(\mathcal{A}) - \Lambda(\mathcal{A}(W_l, S))|$. So, there can be two cases:

1. $\Lambda(\mathcal{A}(W_l, S)) = \Lambda(\mathcal{A}) + \Delta(\mathcal{F}_S)$. Since, $\Lambda(\mathcal{A}) \geq 0$ and $\Delta(\mathcal{F}_S) \geq 0$, $\Lambda(\mathcal{A}(W_l, S)) \geq 0$. So, $\mathcal{F}_S$ is sufficient.

2. $\Lambda(\mathcal{A}(W_l, S)) = \Lambda(\mathcal{A}) - \Delta(\mathcal{F}_S)$
   $\Delta(\mathcal{F}_S) \leq \Lambda(\mathcal{A})$.
   So, $\Lambda(\mathcal{A}(W_l, S)) \geq 0$. So, $\mathcal{F}_S$ is sufficient.

$\square$

**Lemma 4.** *Let, $P_{max}$ denote the maximum of all priorities $P_{ub}(\mathcal{F}_{n_i})$ over $\mathcal{F}$. If $\mathcal{F}_S \subseteq \mathcal{F}$ and $|\mathcal{F}_S| \leq \lfloor \frac{\Lambda(\mathcal{A})}{P_{max}} \rfloor$, then proof feature set $\mathcal{F}_S^c = \mathcal{F} \setminus \mathcal{F}_S$ is sufficient provided $\Lambda(\mathcal{A}) \geq 0$.*

*Proof.*

$$\forall \mathcal{F}_{n_i} \in \mathcal{F}, P_{ub}(\mathcal{F}_{n_i}) \leq P_{max}$$

$$\text{From Lemma 2, } \Delta(\mathcal{F}_S^c) \leq |\mathcal{F}_S| \times P_{max}$$

$$\text{Also, } |\mathcal{F}_S| \leq \lfloor \frac{\Lambda(\mathcal{A})}{P_{max}} \rfloor$$

$$\text{So, } \Delta(\mathcal{F}_S^c) \leq \Lambda(\mathcal{A})$$

$$\text{From Lemma 3, } \mathcal{F}_S^c \text{ is sufficient.}$$

$\square$

**Definition 4.** *Zero proof features set $Z(\mathcal{F})$ denotes the proof features $\mathcal{F}_{n_i} \in \mathcal{F}$ with $P_{ub}(\mathcal{F}_{n_i}) = 0$.*

**Theorem 2.** *Let, $P_{max}$ denote the maximum of all priorities $P_{ub}(\mathcal{F}_{n_i})$ over $\mathcal{F}$. Given any network $N$ is verified on $(\phi, \psi)$ with verifier $\mathcal{V}$ then $|\boldsymbol{F}_{S_0}| \leq d_{l-1} - |Z(\mathcal{F})| - \lfloor \frac{\Lambda(\mathcal{A})}{P_{max}} \rfloor$*

*Proof of theorem 2.* The algorithm 1 arranges the elements of the proof feature set $\mathcal{F}$ in decreasing order according to the priority defined by $P_{ub}$.
Let $\mathcal{F}'$ be the ordered set corresponding to $\mathcal{F}$. So, $\mathcal{F}' = \mathcal{F}_{n_1} :: \cdots :: \mathcal{F}_{n_m}$, where :: is the list concatenation.
The elements of $Z(\mathcal{F})$ will be at the end of this ordering. So, $\mathcal{F}'$ can be written as $\mathcal{F}'' :: Z(\mathcal{F})$ where $Z(\mathcal{F}) = \mathcal{F}_{n_{k+1}} :: \cdots :: \mathcal{F}_{n_m}$ and $\mathcal{F}'' = \mathcal{F}_{n_1} :: \cdots :: \mathcal{F}_{n_k}$ and $p$ be some of the last elements of $\mathcal{F}''$

s.t. the sum of their priorities just less than $\lfloor \frac{\Lambda(\mathcal{A})}{P_{max}} \rfloor$, i.e.,

$$p = \mathcal{F}_{n_j} :: \cdots :: \mathcal{F}_{n_k} \text{ such that}$$

$$\sum_{i=j}^{k} P_{ub}(\mathcal{F}_{n_i}) \leq \Lambda(\mathcal{A})$$

$$\sum_{i=j-1}^{k} P_{ub}(\mathcal{F}_{n_i}) > \Lambda(\mathcal{A})$$

Further, let $p' = p :: Z(\mathcal{F})$, i.e., $p' = \mathcal{F}_{n_j} :: \cdots :: \mathcal{F}_{n_m}$.

Since $P_{ub}$ is 0 for all elements of $Z(\mathcal{F})$,

$$\sum_{i=j}^{m} P_{ub}(\mathcal{F}_{n_i}) \leq \lfloor \frac{\Lambda(\mathcal{A})}{P_{max}} \rfloor \tag{1}$$

Also, in this case, $|p| \geq \lfloor \frac{\Lambda(\mathcal{A})}{P_{max}} \rfloor$ and $|p'| = |p| + |Z(\mathcal{F})|$

We show that $|\boldsymbol{F}_{S_0}| \leq d_{l-1} - |p'|$ which implies $|\boldsymbol{F}_{S_0}| \leq d_{l-1} - \lfloor \frac{\Lambda(\mathcal{A})}{P_{max}} \rfloor - |Z(\mathcal{F})|$.

Now, we prove by induction on the number of steps of the while loop in the algorithm 1 that the set $\boldsymbol{F}_{S_0}$ never contains any elements from $p'$ which implies $|\boldsymbol{F}_{S_0}| \leq d_{l-1} - |p'|$.
**Induction Hypothesis:** $\boldsymbol{F}_{S_0} \cap p' = \{\}$
**Base Case:** At initialization, $\boldsymbol{F}_{S_0} = \{\}$. So, the induction hypothesis holds trivially.
**Induction Step:** Let the induction hypothesis be true for the $n$-th step of the algorithm 1. For the $(n+1)$-th step, let the new $\boldsymbol{F}_{S_0}$ and $\boldsymbol{F}_S$ be $\boldsymbol{F}'_{S_0}$ and $\boldsymbol{F}'_S$ respectively. Consider the following two cases:

1. Let $\boldsymbol{F}_{S_0} \cup \boldsymbol{F}_{S_1}$ be sufficient at line 12. In this case, $\boldsymbol{F}'_{S_0} = \boldsymbol{F}_{S_0}$. So, the induction hypothesis holds.

2. Let $\boldsymbol{F}_{S_0} \cup \boldsymbol{F}_{S_1}$ be not sufficient at line 12.
   **Claim:** $\boldsymbol{F}_{S_0} \cap p' = \{\}$
   Let the above claim be false.
   $\implies \boldsymbol{F}_{S_0} \cap p' \neq \{\}$
   $\implies \mathcal{F} \setminus (\boldsymbol{F}_{S_0} \cup \boldsymbol{F}_{S_1}) \subset p'$
   $\implies \sum\limits_{\mathcal{F}_{n_i} \in \mathcal{F} \setminus (\boldsymbol{F}_{S_0} \cup \boldsymbol{F}_{S_1})} P_{ub} < \lfloor \frac{\Lambda(\mathcal{A})}{P_{max}} \rfloor$  [From (1)]
   $\implies (\boldsymbol{F}_{S_0} \cup \boldsymbol{F}_{S_1})$ is sufficient. (From Lemma 4)
   $\implies$ Contradiction.
   So, $\boldsymbol{F}_{S_1} \cap p' = \{\}$. In this step, $\boldsymbol{F}'_{S_0} = \boldsymbol{F}_{S_0} \cup \boldsymbol{F}_{S_1}$. Also, from induction hypothesis, $\boldsymbol{F}_{S_0} \cap p' = \{\}$. Therefore, the induction hypothesis holds, i.e., $\boldsymbol{F}'_{S_0} \cap p' = \{\}$.

$\square$

# B  VISUALIZATION FOR NON-DIFFERENTIABLE VERIFIER

For non-differentiable verifiers (Katz et al., 2017), we adapt the existing local visualization techniques (Sundararajan et al., 2017; Smilkov et al., 2017) for visualizing the extracted proof features with a sampling-based statistical estimation method. Given a proof feature $\mathcal{F}_{n_i}$, we intend to compute $\mathcal{G}(\mathcal{F}_{n_i}, \phi) = \mathbb{E}_{X \sim \phi} \mathcal{G}(n_i, X)$ which is the mean gradient of the output of $n_i$ w.r.t the inputs from $\phi$. For each input dimension (pixel in case of images) $j \in [d_0]$ the $j$-th component of $\mathcal{G}(\mathcal{F}_{n_i}, \phi)$ estimates its relevance w.r.t proof feature $\mathcal{F}_{n_i}$ - with higher gradient values representing higher relevance. Given that the input region $\phi$ contains infinitely many inputs, instead of exactly computing $\mathcal{G}(\mathcal{F}_{n_i}, \phi)$ we statistically estimate it by a reasonably large sample drawn uniformly from $\phi$.

# C  ADDITIONAL EXPERIMENTS

## C.1  DISTRIBUTION PLOTS OF THE SIZE OF THE EXTRACTED SUFFICIENT PROOF FEATURE SET

In this section, we show plots for the distribution of the sufficient proof feature set size extracted by ProFIt and two baseline methods based on random and gradient-based priority heuristics. In the following histograms, the x-axis represents the size of the extracted proof feature set and the y-axis represents the number of local robustness properties. The results show that ProFIt consistently outperforms both the baselines and extracts sufficient proof feature sets that are smaller in size. All the plots in this section are generated on 500 local robustness properties with $\alpha$-Crown verifier.

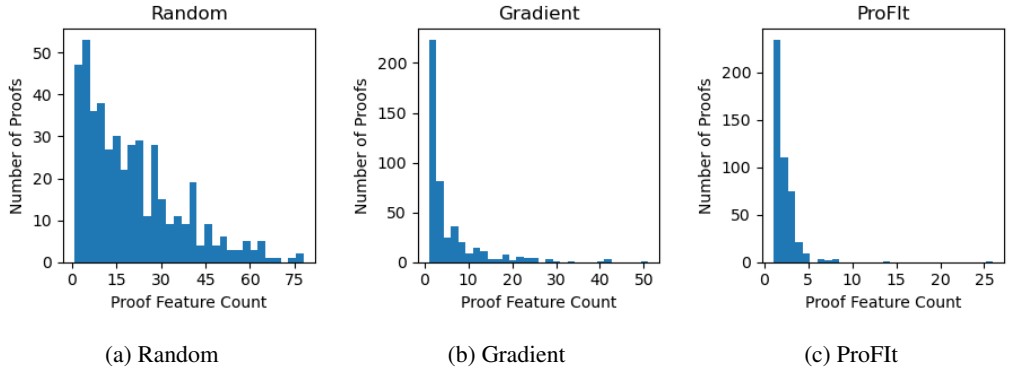

(a) Random          (b) Gradient          (c) ProFIt

Figure 4: Distribution of the extracted proof feature set size - Standard MNIST network & $\epsilon = 0.02$.

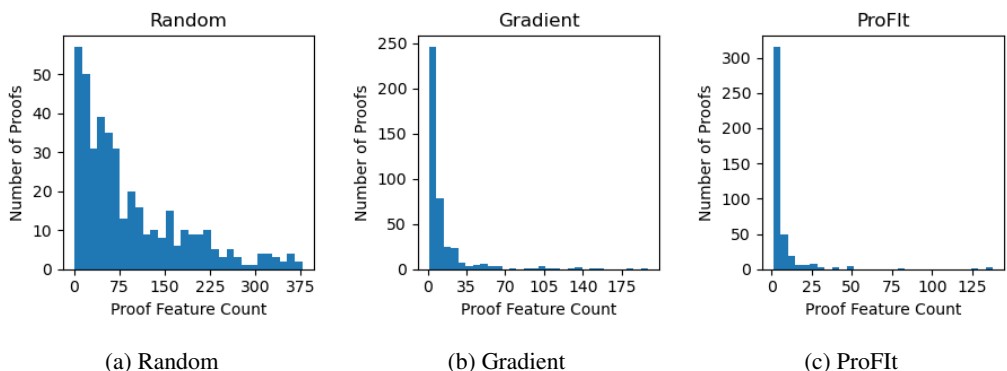

(a) Random          (b) Gradient          (c) ProFIt

Figure 5: Distribution of the extracted proof feature set size - PGD MNIST network & $\epsilon = 0.02$.

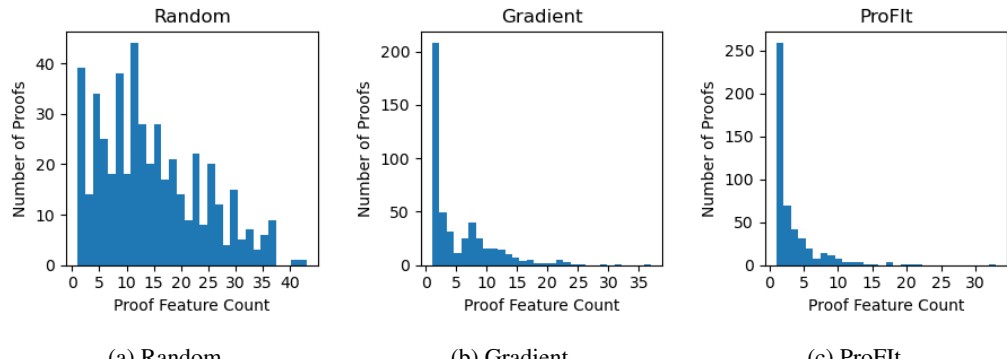

(a) Random            (b) Gradient            (c) ProFIt

Figure 6: Distribution of the extracted proof feature set size - Colt MNIST network & $\epsilon = 0.02$.

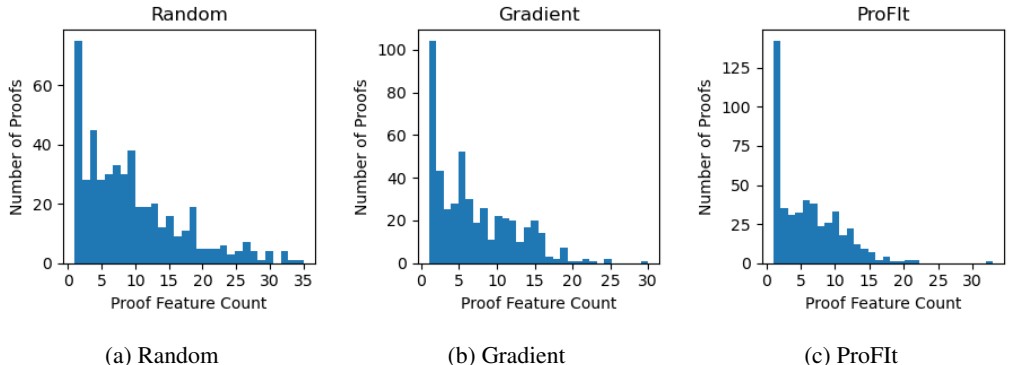

(a) Random            (b) Gradient            (c) ProFIt

Figure 7: Distribution of the extracted proof feature set size - Crown-IBP MNIST network & $\epsilon = 0.02$.

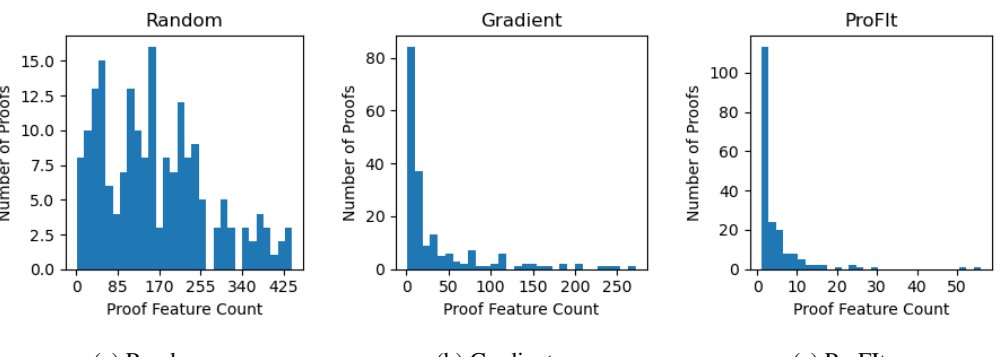

(a) Random            (b) Gradient            (c) ProFIt

Figure 8: Distribution of the extracted proof feature set size - PGD MNIST network & $\epsilon = 0.1$.

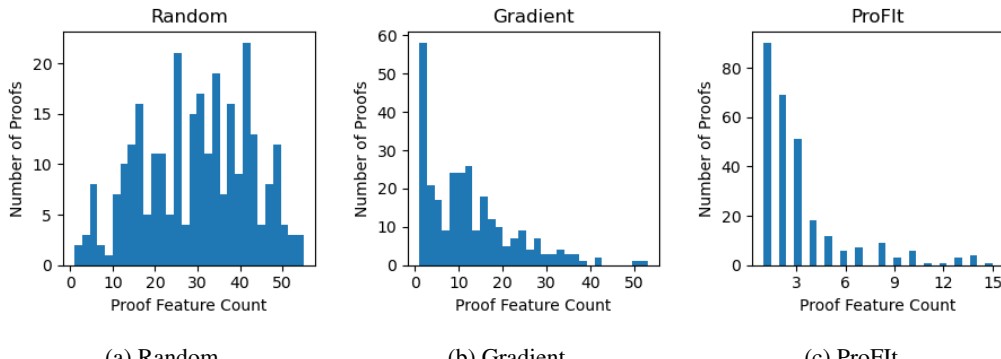

(a) Random  (b) Gradient  (c) ProFIt

Figure 9: Distribution of the extracted proof feature set size - Colt MNIST network & $\epsilon = 0.1$.

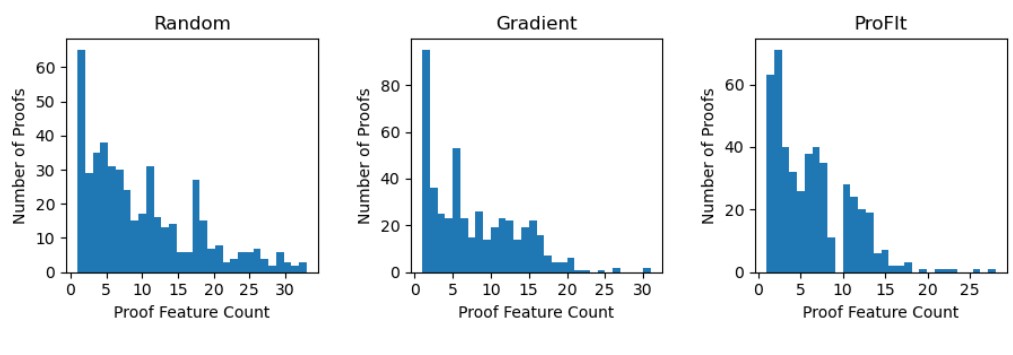

(a) Random  (b) Gradient  (c) ProFIt

Figure 10: Distribution of the extracted proof feature set size - Crown-IBP MNIST network & $\epsilon = 0.1$.

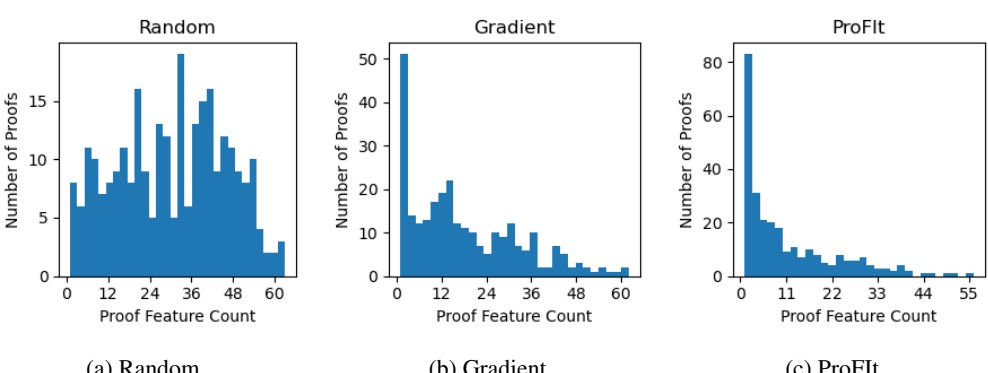

(a) Random  (b) Gradient  (c) ProFIt

Figure 11: Distribution of the extracted proof feature set size - Standard Cifar-10 network & $\epsilon = 0.2/255$.

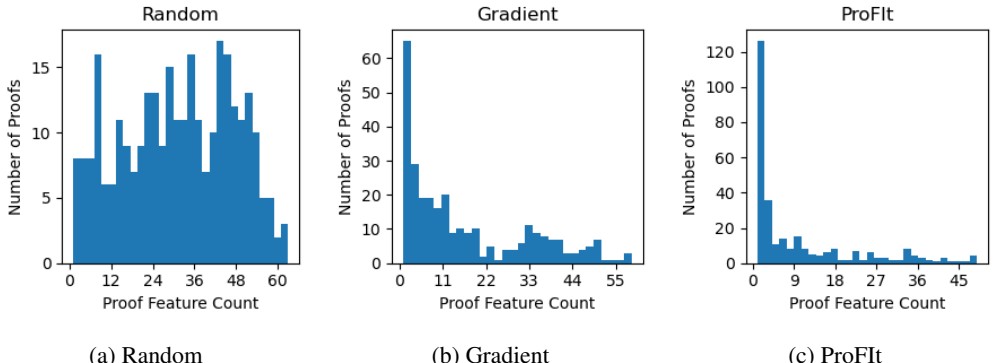

(a) Random            (b) Gradient            (c) ProFIt

Figure 12: Distribution of the extracted proof feature set size - PGD Cifar-10 network & $\epsilon = 0.2/255$.

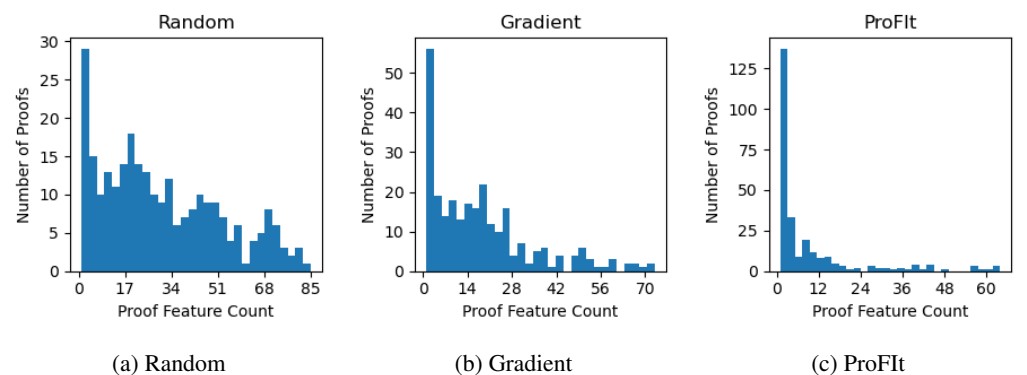

(a) Random            (b) Gradient            (c) ProFIt

Figure 13: Distribution of the extracted proof feature set size - Colt Cifar-10 network & $\epsilon = 0.2/255$.

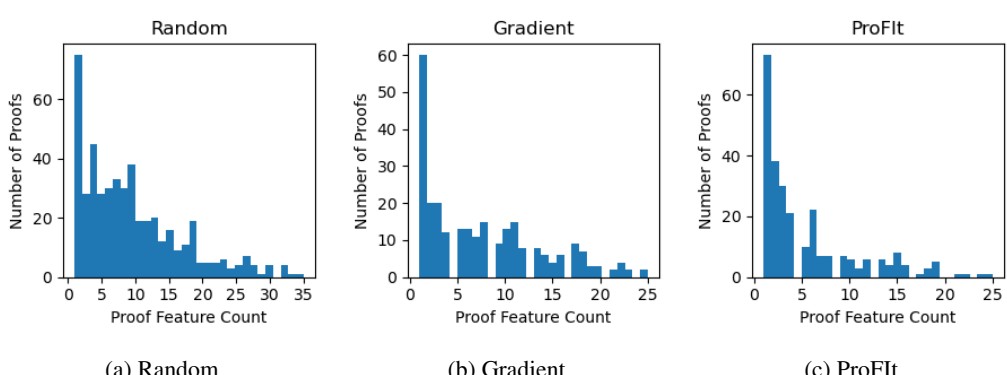

(a) Random            (b) Gradient            (c) ProFIt

Figure 14: Distribution of the extracted proof feature set size - Crown-IBP Cifar-10 network & $\epsilon = 0.2/255$.

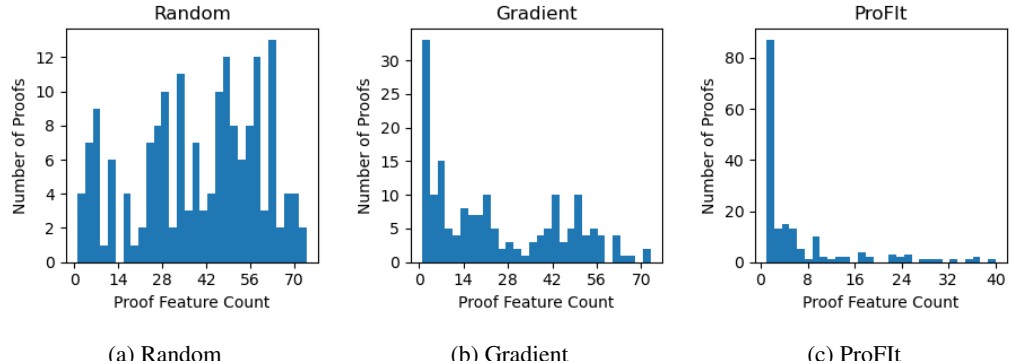

(a) Random          (b) Gradient          (c) ProFIt

Figure 15: Distribution of the extracted proof feature set size - PGD Cifar-10 network & $\epsilon = 2/255$.

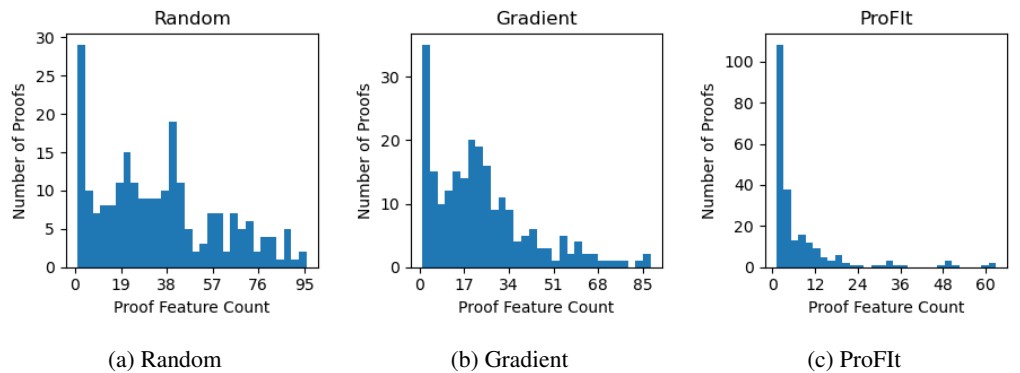

(a) Random          (b) Gradient          (c) ProFIt

Figure 16: Distribution of the extracted proof feature set size - Colt Cifar-10 network & $\epsilon = 2/255$.

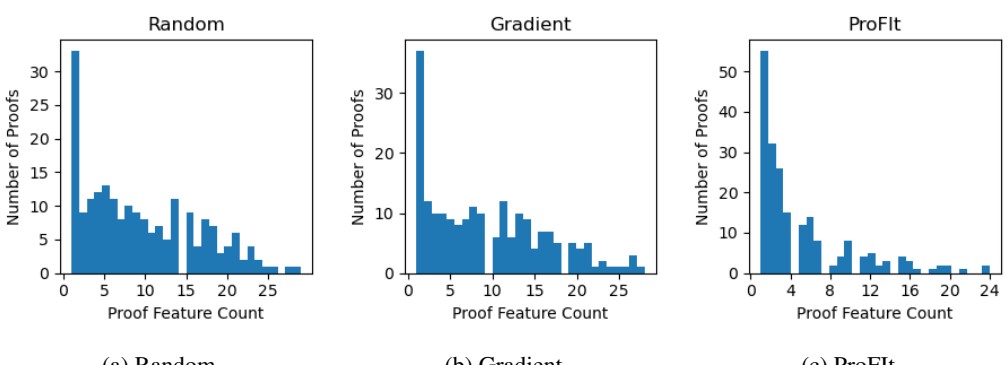

(a) Random          (b) Gradient          (c) ProFIt

Figure 17: Distribution of the extracted proof feature set size - Crown-IBP Cifar-10 network & $\epsilon = 2/255$.

## C.2 ADDITIONAL PLOTS FOR PRIORITY ORDER EVALUATION

In this section, we evaluate the efficacy of the priority ordering of proof features defined in Eq. 3 against the random and gradient-based priority ordering on Cifar-10 networks and the standard MNIST network. We use $\epsilon = 0.2/255$ for all Cifar-10 networks and $\epsilon = 0.02$ for the standard MNIST network. We show that the priority ordering used by ProFIt preserves a higher % of proofs while better approximating the original verifier output i.e. achieving a lower relative change compared to both the baselines. All the plots in this section generated on 500 local properties with $\alpha$-Crown verifier.

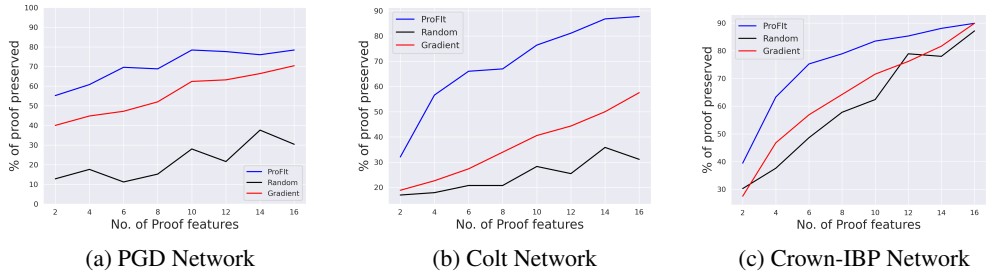

(a) PGD Network      (b) Colt Network      (c) Crown-IBP Network

Figure 18: Percentages of proofs preserved by different heuristics on robust CIFAR-10 networks.

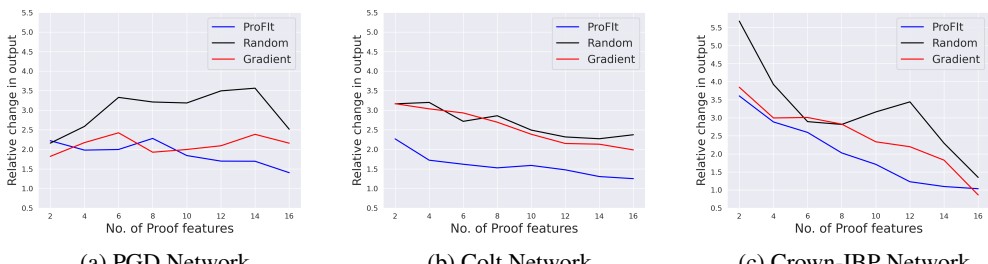

(a) PGD Network      (b) Colt Network      (c) Crown-IBP Network

Figure 19: Relative change in verifier output with different heuristics on robust CIFAR-10 networks.

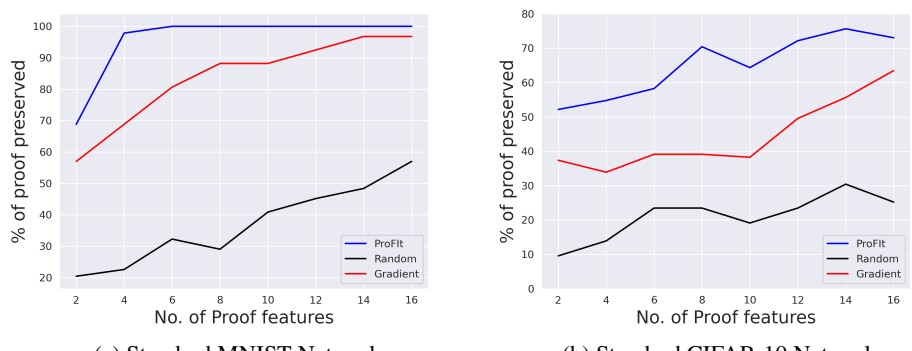

(a) Standard MNIST Network         (b) Standard CIFAR-10 Network

Figure 20: Percentages of proofs preserved by different heuristics on standard networks.

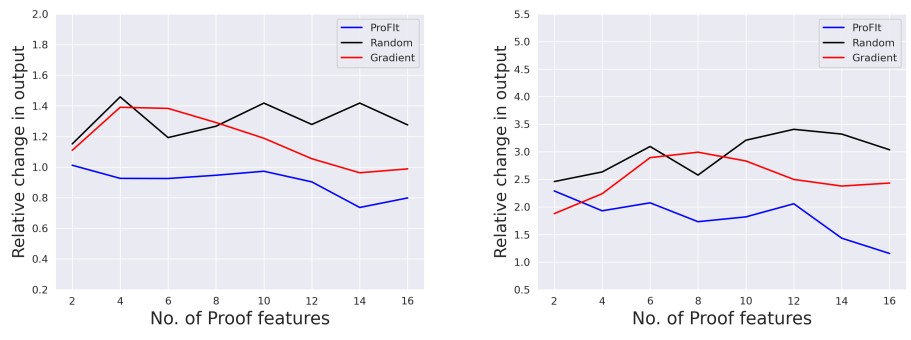

(a) Standard MNIST Network         (b) Standard CIFAR-10 Network

Figure 21: Relative change in verifier output with different heuristics on standard networks.

### C.3 QUALITATIVE EVALUATION OF THE PRIORITY ORDERING OF THE PROOF FEATURES

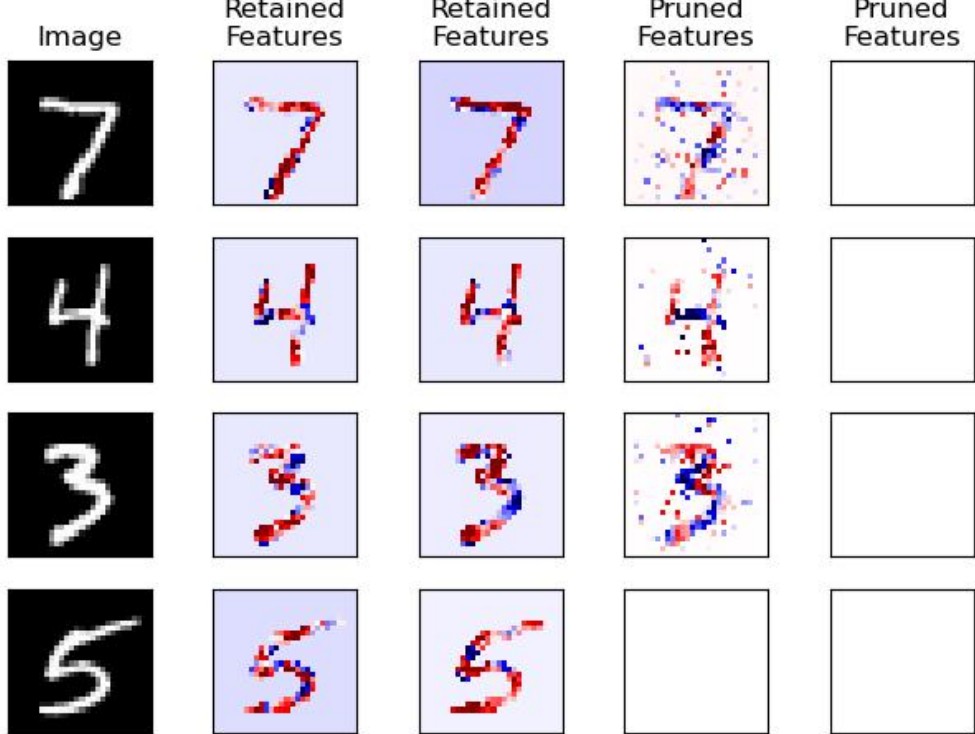

Figure 22: Comparing gradients of the top proof features retained by the ProFIt algorithm to pruned proof features with a low priority. The columns are sorted in decreasing order of priority with the right-most proof feature having the lowest priority. As expected, proof features with low priority are noisier (as in column 4). The proof features which are further down in the priority order do not give any relevant information about the input (column 5 - completely white). This shows that the proposed algorithm extracts proof features that are important to the proof while removing insignificant and uninformative proof features. The shown gradients are computed on COLT-trained MNIST network.

## C.4 ADDITIONAL PLOTS FOR THE TOP PROOF FEATURE VISUALIZATION

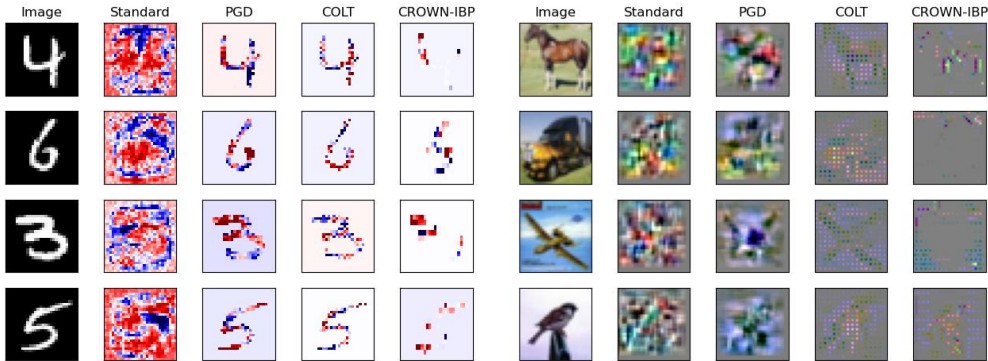

(a) Gradient maps generated on MNIST networks     (b) Gradient maps generated on CIFAR-10 networks

Figure 23: Additional plots for the top proof feature visualization (in addition to Fig. 3) - Visualization of gradient map of top proof feature (having highest priority) generated for networks trained with different training methods. It is evident that the top proof feature corresponding to the standard network highlights both relevant and spurious input features. In contrast, the top proof feature of the provably robust network does filter out the spurious input features, but it comes at the expanse of some important input features. The top proof features of the networks trained with PGD filter out more spurious features as compared to standard networks. Finally, the top proof features of the networks trained with COLT filter out the spurious input features and also correctly highlight the relevant input features.

C.5    VISUALIZATION OF THE ENTIRE EXTRACTED PROOF FEATURE SET

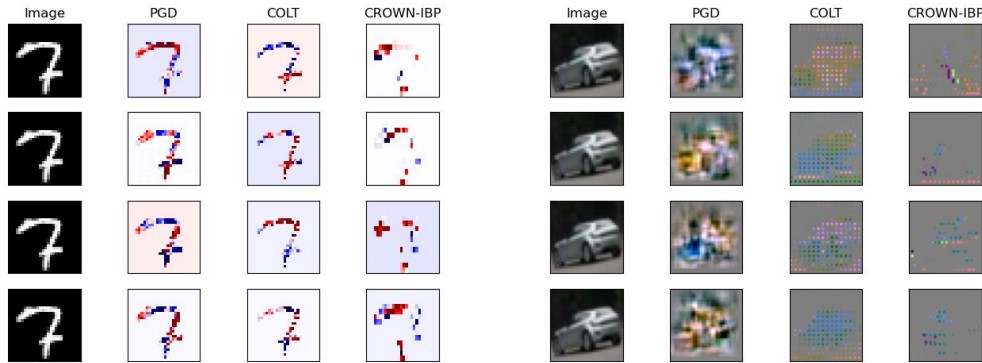

(a) Gradient maps generated on MNIST networks          (b) Gradient maps generated on CIFAR-10 networks

Figure 24: Visualization of gradient maps of entire proof features set consisting of 4 proof features extracted for networks trained with different robust training methods. The gradient maps of the proof features are presented in decreasing order of priority with the top row showing the gradient map corresponding to the top proof feature of each network.

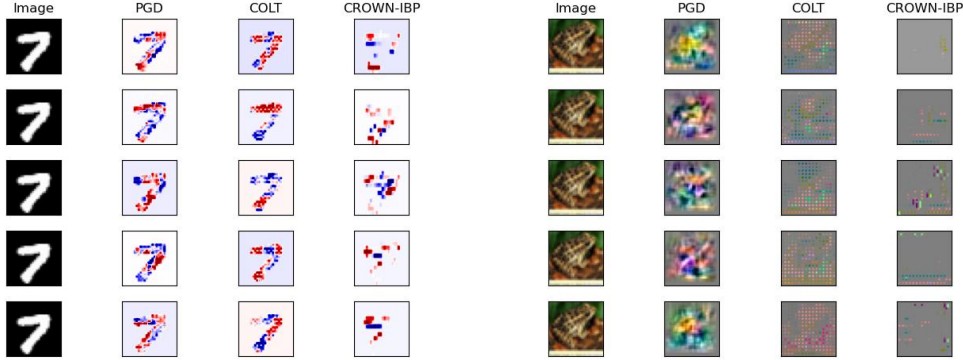

(a) Gradient maps generated on MNIST networks          (b) Gradient maps generated on CIFAR-10 networks

Figure 25: Visualization of gradient maps of entire proof features set consisting of 5 proof features extracted for networks trained with different robust training methods. The gradient maps of the proof features are presented in decreasing order of priority with the top row showing the gradient map corresponding to the top proof feature of each network.

## C.6 VISUALIZATION OF THE TOP PROOF FEATURE FOR HIGHER $\epsilon$ VALUES

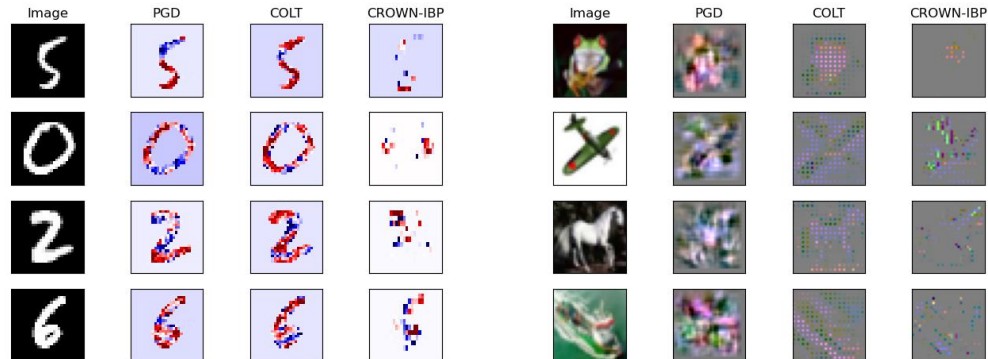

(a) Gradient maps generated on MNIST networks    (b) Gradient maps generated on CIFAR-10 networks

Figure 26: Visualization of gradient map of top proof feature (having highest priority) generated for networks trained with different robust training methods. For these networks, we define local properties with higher $\epsilon$ values. For MNIST networks and CIFAR-10 networks, we take $\epsilon = 0.1$ and $\epsilon = 2/255$ respectively.

## C.7 COMPARING THE PROOF FEATURES EXTRACTED BY DIFFERENT VERIFIERS

As described earlier, the extracted proof features are specific to the proof generated by a particular verifier. In this section, we examine whether the proof features corresponding to different proofs generated by different verifiers are the same. In Table 2, we compare the top proof features (ordered by the priority order described in Eq. 3) of the proofs generated by the DeepZ Singh et al. (2018a) verifier and $\alpha$-Crown verifier. We observed that more than $97\%$ of the cases the top feature remains the same. In this case, for different verifiers though the proof features $\mathcal{F}_{n_i} = [l_{n_i}, u_{n_i}]$ change, their relative priority ordering computed by Eq. 3 remains the same. For this experiment, we use 500 local robustness properties for each network and each $\epsilon$ value. Note, while comparing proof features from two different verifiers we only consider those properties that can be proved by both the verifiers.

| Dataset | Training Method | Input Region ($\phi$) eps ($\epsilon$) | % properties proved by DeepZ | % properties proved by $\alpha$-Crown | % proofs with the same top feature | % proofs with the same top-5 feature | % proofs with the same top-10 feature |
|---------|-----------------|------------|------|------|------|------|------|
| MNIST | Standard | 0.02 | 90.0 % | 91.8 % | 99.8 % | 98.4 % | 98.3 % |
| | PGD Trained | 0.02 | 82.0 % | 83.0 % | 99.75 % | 99.0 % | 98.0 % |
| | COLT | 0.02 | 95.4 % | 96 % | 99.50 % | 98.95 % | 96.25 % |
| | CROWN-IBP | 0.02 | 96.4 % | 96.4 % | 99.8 % | 99.0 % | 95.9 % |
| MNIST | PGD Trained | 0.1 | 32.6 % | 38.2 % | 99.38 % | 95.7 % | 91.41 % |
| | COLT | 0.1 | 43.0 % | 56.2 % | 98.6 % | 93.95 % | 87.90 % |
| | CROWN-IBP | 0.1 | 89.4 % | 94.6 % | 97.0 % | 88.3 % | 80.26 % |
| CIFAR-10 | Standard | 0.2/255 | 51.0 % | 58.0 % | 99.5 % | 98.3 % | 98.0 % |
| | PGD Trained | 0.2/255 | 47.0 % | 62.5 % | 99.7 % | 98.5 % | 97.8 % |
| | COLT | 0.2/255 | 53.0 % | 53.0 % | 100.0 % | 99.5 % | 98.2 % |
| | CROWN-IBP | 0.2/255 | 54.5 % | 54.5 % | 100.0 % | 98.90 % | 97.8 % |
| CIFAR-10 | PGD Trained | 2/255 | 26.5 % | 32.5 % | 99.7 % | 95.45 % | 92.45 % |
| | COLT | 2/255 | 45.5 % | 46.0 % | 99.8 % | 95.9 % | 97.3 % |
| | CROWN-IBP | 2/255 | 37.5 % | 38.0 % | 99.6 % | 97.92 % | 95.89 % |

Table 2: Comparing extracted proof features of DeepZ & $\alpha$-Crown

Next, in Table 3 we compare the top proof feature (having the highest priority) corresponding to the proofs generated by DeepZ, Crown (Zhang et al., 2018), $\alpha$-Crown and state-of-the-art complete verifier $\alpha, \beta$-Crown (Wang et al., 2021b) on the property. We use PGD and Colt MNIST network for this experiment. We evaluate 200 local robustness properties defined with $\epsilon = 0.02$. We observed that for these two networks, more than $99\%$ of the cases the top feature remains the same.

| Verifier | DeepZ | Crown | $\alpha$-Crown | $\alpha, \beta$-Crown | Verifier | DeepZ | Crown | $\alpha$-Crown | $\alpha, \beta$-Crown |
|----------|-------|-------|---------|----------|----------|-------|-------|---------|----------|
| DeepZ | 100.0 % | 99.55 % | 99.75 % | 99.50 % | DeepZ | 100.0 % | 99.50 % | 99.50 % | 99.40 % |
| Crown | 99.55 % | 100.0 % | 99.80 % | 99.60 % | Crown | 99.50 % | 100.0 % | 100.0 % | 99.70 % |
| $\alpha$-Crown | 99.75 % | 99.80 % | 100.0 % | 99.80 % | $\alpha$-Crown | 99.50 % | 100.0 % | 100.0 % | 99.70 % |
| $\alpha, \beta$-Crown | 99.50 % | 99.60 % | 99.80 % | 100.0 % | $\alpha, \beta$-Crown | 99.40 % | 99.70 % | 99.70 % | 100.0 % |

(a) PGD MNIST Network     (b) Colt MNIST Network
Table 3: % cases different verifiers have the same top proof feature

## C.8   COMPARING VISUALIZATION OF THE PROOF FEATURES EXTRACTED BY DIFFERENT DIFFERENTIABLE VERIFIERS

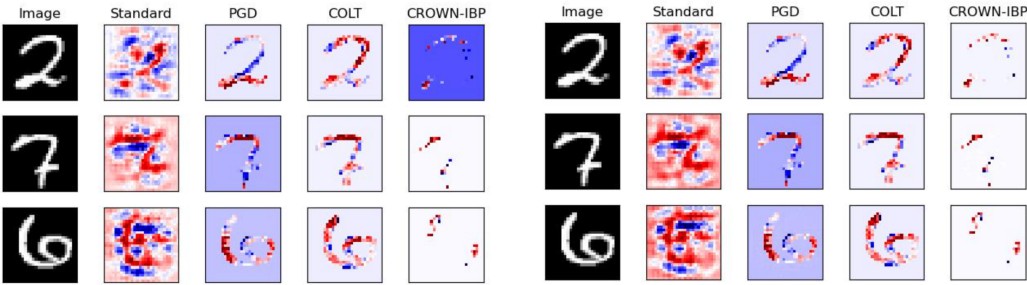

(a) Gradient maps generated with CROWN          (b) Gradient maps generated with $\alpha$-CROWN

Figure 27: Visualization of gradient map of top proof feature extracted by two different differentiable verifiers - CROWN and $\alpha$-CROWN. In both cases, the visualizations align with the observations in section 5.3.

## C.9    COMPARING PROOFS ON NETWORKS WITH SAME ARCHITECTURE

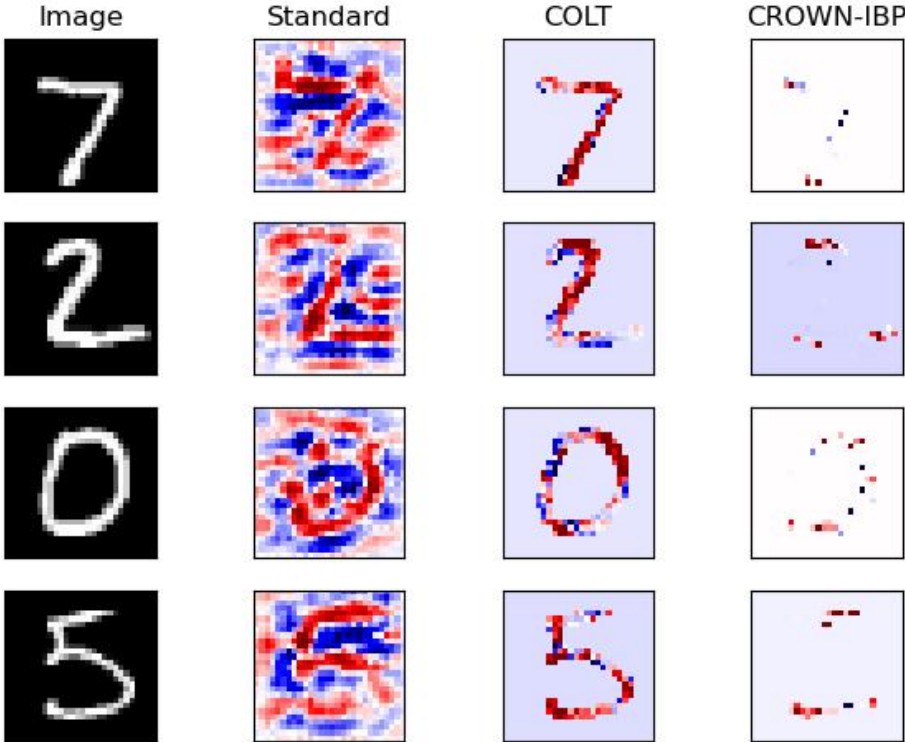

Figure 28: Gradient maps generated on MNIST networks trained with different training methods (Standard, COLT, CROWN-IBP) with the same architecture. The gradient maps show that observations in Section 5.3 and in Figure 3 of the paper also hold on different networks with the same architecture.

## C.10 Plots for sensitivity analysis w.r.t $\epsilon_{train}$

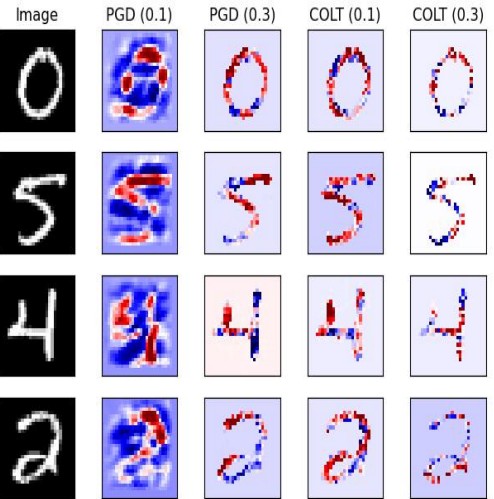

Figure 29: Plots for visualizing gradients of the top proof feature for PGD and COLT networks trained using different values of $\epsilon_{train} \in \{0.1, 0.3\}$ The gradient map corresponding to the networks trained with the higher value of $\epsilon_{train}$ filter out more input features than the ones trained with smaller $\epsilon_{train}$ value.

### C.11 Plots for the top proof feature visualization of SABR networks

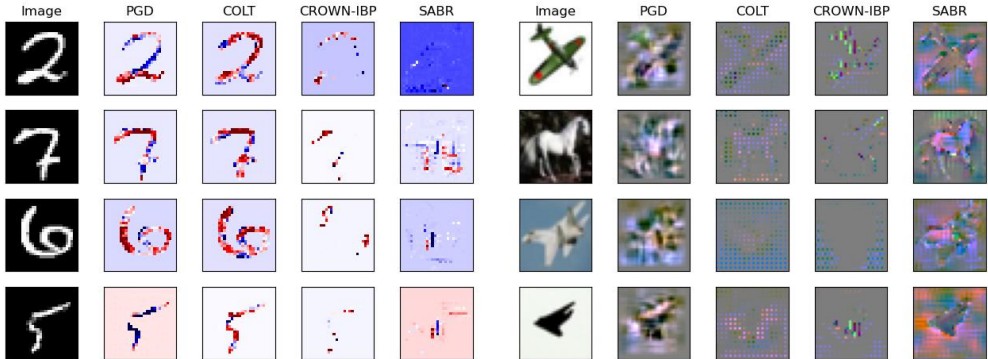

(a) Gradient maps generated on MNIST networks  (b) Gradient maps generated on CIFAR-10 networks

Figure 30: Visualization of gradient maps of the top proof feature (having highest priority) generated for networks including those trained with recent state-of-the-art training method SABR (Mueller et al., 2023). For the SABR CIFAR10 network, the gradient map of the top feature highlights semantically meaningful features while successfully removing spurious background pixels. However, for the SABR MNIST network, the gradient map of the top proof feature is not human-aligned. This is potentially because the SABR MNIST networks are over-regularized to achieve high robust accuracy.

## D  Worst-case runtime analysis of ProFIt

In this section, We provide a formal worst-case analysis of the ProFIt.

**Theorem 3.** *If the size of the penultimate layer is $d_{l-1}$ then the number of incremental verifier calls made by ProFIt is always $\leq \lceil \log d_{l-1} \rceil + 1$.*

*Proof.* Every iteration of the while loop (Algorithm 1 on page 6 of the paper) makes a single incremental verifier call. So, first, we calculate the maximum number of incremental verifier calls made in the while loop. At the end of each iteration $\boldsymbol{F}_S$ is replaced by either $\boldsymbol{F}_{S_1}$ (line 14 of Algorithm 1) or $\boldsymbol{F}_{S_2}$ (line 17 of Algorithm 1). Now, $\max(|\boldsymbol{F}_{S_1}|, |\boldsymbol{F}_{S_2}|) \leq (|\boldsymbol{F}_S| + 1)/2$ where $|\cdot|$ denotes the cardinality of a set. Initially, $\boldsymbol{F}_S = \mathcal{F}$ and $|\boldsymbol{F}_S| = |\mathcal{F}| = d_{l-1}$. Then after $i$th iteration of the while loop the following inequality holds $|\boldsymbol{F}_S| \leq \frac{d_{l-1}}{2^i} + \sum_{j=1}^{i} \frac{1}{2^j}$. Let, $I$ denote the total number of iterations of the while loop. Given the while loop only executes when $\boldsymbol{F}_S$ is non-empty (line 9 of Algorithm 1), $1 \leq \frac{d_{l-1}}{2^{(I-1)}} + \sum_{j=1}^{I-1} \frac{1}{2^j} \implies (I-1) \leq \lceil \log d_{l-1} \rceil \implies I \leq \lceil \log d_{l-1} \rceil + 1$.

Hence, the total number of incremental verifier calls is always $\leq \lceil \log d_{l-1} \rceil + 1$. Note, since we only modify the final layer we run incremental verification that avoids rerunning verification on the entire network from scratch. However, computing proof features (line 6 of Algorithm 1) requires a single verifier call which can not be run incrementally.

For a network with $n$ neurons and $l$ layers a single incremental DeepZ (Singh et al., 2018a) verifier call in the worst case takes $O(n^3)$ time whereas a single call to incremental CROWN (Zhang et al., 2018) verifier in the worst case takes $O(l \times n^3)$ time. The cost for a single non-incremental DeepZ verifier call is in the worst case $O(l \times n^3)$ whereas for CROWN it is $O(l^2 \times n^3)$. Overall, the complexity of ProFIt with CROWN is $O((\log d_{l-1} + l) \times l \times n^3)$. We will update our paper with detailed proof of the worst-case runtime of ProFIt. $\square$

| Dataset | Network | Training | # Layers | # Params | Theoretical Bound | Avg. verifier calls | Avg. CPU time | Avg. GPU time |
|---------|---------|----------|----------|----------|-------------------|---------------------|---------------|---------------|
| **MNIST** | ConvSmall | Standard | 4 | 90K | 9 | 7.81 | 0.85s | 0.14s |
| | ConvMed | PGD | 5 | 200K | 12 | 9.90 | 1.80s | 0.16s |
| | ConvSmall | COLT | 4 | 90K | 9 | 7.25 | 0.39s | 0.11s |
| | IBP-Small | CROWN-IBP | 4 | 80K | 9 | 6.99 | 0.34s | 0.09s |
| | ConvBig | DiffAI | 7 | 1.8M | 11 | 7.12 | 2.57s | 0.29s |
| | ConvSuper | DiffAI | 7 | 10M | 11 | 7.08 | 6.28s | 0.30s |
| **CIFAR10** | ConvSmall | Standard | 4 | 120K | 9 | 5.36 | 0.30s | 0.07s |
| | ConvSmall | PGD | 4 | 120K | 9 | 6.04 | 0.32s | 0.09s |
| | ConvSmall | COLT | 4 | 120K | 10 | 5.64 | 0.52s | 0.15s |
| | IBP-Small | CROWN-IBP | 4 | 100K | 10 | 6.2 | 0.48s | 0.11s |
| | ConvBig | DiffAI | 7 | 2.5M | 11 | 7.8 | 3.93s | 0.50s |

Table 4: ProFIt Runtime Analysis

# E    EXPERIMENTAL RESULTS ON PROFIT RUNTIME

## E.1    EXPERIMENT SETUP

We run ProFIt on multiple networks including those mentioned in Table 1 on both CPUs and GPUs. For CPU-related experiments, we use the same setup as mentioned in Section 5.1. In GPU-related experiments, we utilize a single NVIDIA A100-PCI GPU with 40 GB RAM. We use the state-of-the-art incomplete verifier $\alpha$-CROWN from auto-LiRPA (Xu et al., 2020) toolbox and show results for 100 local $L_\infty$ robustness properties. As done in the paper, for MNIST and CIFAR10 networks, we use $\epsilon = 0.02$ and $\epsilon = 0.2/255$ respectively to define $L_\infty$ input regions.

## E.2    SIZE AND ARCHITECTURE OF THE NETWORKS:

We present the runtime analysis of ProFIt on the DNNs used in Table 1 and also showcase results on some of the largest verifiable network architectures, namely DiffAi trained (Mirman et al., 2018) ConvBig and ConvSuper. These architectures are de facto benchmarks employed for testing the scalability of state-of-the-art verifiers (Ferrari et al., 2022b; Müller et al., 2022) and are featured in the International Verification of Neural Networks Competition (Brix et al., 2023). All the networks used in this experiment are selected from the ERAN (Singh et al., 2019b), and CROWN-IBP (Zhang et al., 2020) repositories and belong to the category of convolutional neural networks (CNNs).

## E.3    COMPATIBILITY WITH GPU ACCELERATION:

ProFIt is not restricted to CPUs and works with DNN verifiers whose computation can benefit from GPU acceleration. For example, ProFIt can be run with GPU implementation of auto-LiRPA (Xu et al., 2020). We show the runtime improvement of ProFIt with GPUs below. Runtimes on GPUs are up to 20x shorter than those on CPUs.

## E.4    EXPERIMENTAL RESULTS:

We present results for all networks in Table 4. Column 1 shows the dataset, column 2 displays the network name, column 3 outlines the training method, and columns 4 and 5 describe the network's structure. Column 6 indicates the worst-case bound on the number of incremental verifier calls, as described in Theorem 3, while column 7 displays the average number of incremental verifier calls per property. Columns 8 and 9 present the average runtime in seconds for ProFIt on CPU and GPU, respectively. Notably, on GPUs, the average runtime for ProFIt is less than 1 second for all the networks. Note that, for ConvBig architectures, the timeout used in the International Verification of Neural Networks Competition (Brix et al., 2023) is in minutes. In comparison, ProFIt with $\alpha$-CROWN is significantly faster.

