# OpenReview forum: "Interpreting Robustness Proofs of Deep Neural Networks"
_ICLR.cc/2024/Conference — ICLR 2024 poster_

### Official Review · Reviewer_UvmS · 2023-10-16

**Soundness:** 3 good
**Presentation:** 2 fair
**Contribution:** 2 fair
**Rating:** 6
**Confidence:** 3

**Summary:**

This paper proposes the method ProFIt to extract a set of proof features by retaining only the more important parts of the proof that preserves the property and then conducts experiments to explore the selected features for standard trained / adversarial trained / certifiable trained neural networks.

**Strengths:**

- The proposed method is novel and theoretically guaranteed.
- The experiments validate the sufficiency of the proposed method.

**Weaknesses:**

- The proposed algorithm seems only valid for the linear property, how can we deal with the nonlinear properties?
- My main concern is the meaning of this task.
  - The selected features are the most relevant to the predefined linear property. However, there are many other properties of a neural network, different properties lead to different sufficient proof features. So, the features selected by one property may only reflect one perspective of the neural network. Such a "local" interpretation may be misleading.
  - Can sufficient proof features provide intuitions on how to design a training method to train a model with such a property? I think this need to be discussed.

**Questions:**

See the weaknesses.

---

> ### Author Response · Authors · 2023-11-16
>
> **Q1. how can we deal with the nonlinear properties?**
>
> **R1:** Existing works [1, 2] that handle non-linear properties typically compute a linear approximation of these properties and utilize existing DNN verifiers designed for linear specifications. For instance, in [1], when verifying robustness against geometric perturbations like rotation, an overapproximated convex input region is computed, specified by a set of linear inequalities encompassing all rotated images. Subsequently, existing DNN verifiers are applied. A similar approach is employed in [2] for non-linear output specifications.
> Both these cases with linear approximations of non-linear input and output specification can be handled with ProFIt. In general, for DNN verification, nonlinear specifications are rare, and most of the common norm-based local robustness properties can be specified as a conjunction of linear inequalities, and the corresponding proofs can be analyzed with ProFIt.
>
> [1] "Certifying Geometric Robustness of Neural Networks", M. Balunovic, NeurIPS, 2019. \
> [2] "Verification of Non-Linear Specifications for Neural Networks", C. Qin, ICLR, 2019.
>
> **Q2. Different properties lead to different sufficient proof features. So, the features selected by one property may only reflect one perspective of the neural network.**
>
> **R2:** ProFIt analyzes and generates a human-understandable explanation of the proof of a specific property. Different properties capture different behaviors of the DNN and therefore, the proof and subsequently human-understandable explanation generated by ProFIt will reflect these differences. To elaborate, for robustness we want the DNN output to remain correct when input changes. In contrast, for monotonicity we want the output of DNN to change in a certain manner with respect to the change in input. So, it is natural that the proof of different properties on the same DNN will rely on different proof features. In this paper, we focus on proof dissection of the common local robustness properties and leave the dissection of proofs of other properties such as monotonicity to future work.
>
> **Q3. Can sufficient proof features provide intuitions on how to design a training method to train a model with such a property?**
>
> **R3:** Please refer to the response to Q2 in the common response.

---

> > ### Comment · Reviewer_UvmS · 2023-11-16
> >
> > Thank you for your reply. I have no further questions.

---

### Official Review · Reviewer_YfNd · 2023-10-24

**Soundness:** 3 good
**Presentation:** 3 good
**Contribution:** 2 fair
**Rating:** 6
**Confidence:** 3

**Summary:**

The authors consider the problem of making proofs of neural network robustness more human understandable. In particular, the authors consider recent verification methods developed for computing neural networks robustness and extract proof features from these methods by analysing the ranges of the outputs of each neuron at the penultimate layer. Then, they proceed by finding a small subset of these features that are sufficient to explain the verification result. The authors show the effectiveness of their results on experiments on the MNIST and CIFAR datasets.

**Strengths:**

- The paper is well written

- The problem of analysing and explain the decisions of the various methods developed to verify neural networks is of interest to the ICLR community and to the best of my knowledge novel

- The algorithm is sound and experimental results seem to show improvement compared to state of the art

**Weaknesses:**

- The experimental setting is not totally clear to me:

1) how is the original feature count in Table 1 computed? Should not this be just the size of the penultimate layer? Also why is this 10 times larger for PGD trained networks on MNIST compared to PGD trained networks on CIFAR?

2) As ProFIt has a  subroutine that needs to check for sufficiency of a feature set, It would be good to perform experiment to analyse the scalability of the proposed method on various architectures for the various robustness methods considered in the paper (and to report computation times).

- The computational complexity of ProFIt should be discussed more clearly.

**Questions:**

- Can the method also be applied to other certification methods such as randomised smoothing [Cohen, Jeremy, Elan Rosenfeld, and Zico Kolter. "Certified adversarial robustness via randomized smoothing." international conference on machine learning. PMLR, 2019.]?

---

> ### Author Response · Authors · 2023-11-16
>
> **Q1. how is the original feature count in Table 1 computed? Why the size of the penultimate is different for MNIST and CIFAR10 PGD-trained networks?**
>
> **R1:** As pointed out, the original feature count is the same as the size of the penultimate layer. In Table 1, we opted for publicly available state-of-the-art pre-trained networks from official repositories [1, 2, 3] to prevent training networks ourselves and introducing unintended bias in the process. Therefore, the networks in Table 1 do not always share the same architecture and may have different sizes at the penultimate layer. The experiments on networks with the **same architecture** are in Appendix C.9. \
> [1] ERAN - https://github.com/eth-sri/eran \
> [2] CROWN-IBP - https://github.com/huanzhang12/CROWN-IBP \
> [3] COLT - https://github.com/eth-sri/colt \
>
> **Q2. Scalability analysis of ProFIt**
>
> **R2:** Please refer to the response to Q1 in common response.
>
> **Q3. The computational complexity of ProFIt**
>
> **R2:** Please refer to the response to Q1 in common response.
>
> **Q4. Can the method also be applied to probabilistic certification methods such as randomised smoothing?**
>
> **R4:** ProFIt is designed for deterministic DNN verifiers and currently cannot handle probabilistic verifiers like randomized smoothing. To extend ProFIt to accommodate probabilistic verifiers, we would first need to define the priority ordering of the proof features w.r.t the proof generated by probabilistic verifiers. Additionally, determining how to generate visualizations of the proof features poses a challenge since, unlike common incomplete verifiers, the computation of randomized smoothing cannot be represented as differentiable programs. Both of these aspects require further investigation, and we leave them as subjects for future work. We will update Section 4.5 of the paper to include this limitation.

---

> > ### Author Response · Authors · 2023-11-21
> >
> > Dear reviewer YfNd,
> >
> > As the discussion period is coming to a close, we would like to take the opportunity to thank you for your time once again. We hope our comments so far have addressed all prior concerns/questions, but please let us know if there is anything we can elaborate on. If you have any other experiments or questions, please let us know and we will do our best to run/address anything before the deadline.

---

### Official Review · Reviewer_zRr5 · 2023-10-31

**Soundness:** 3 good
**Presentation:** 3 good
**Contribution:** 3 good
**Rating:** 6
**Confidence:** 2

**Summary:**

This paper proposes to verify the robustness of neural classifiers by finding the "robust features" via pruning the penultimate layer features based on their "priority". The resulting "ProFIt" algorithm guarantees sufficiency and reduces the number of proof features required to certify an area around the input, reducing the complexity of the certificate. The authors also show that high-priority robust features can visualize the neural classifier attention and use such a property to compare various robust models.

**Strengths:**

The paper proposes a theoretically-inspired heuristic algorithm that seems to work as expected. The algorithm provides improved interpretability to the robust verification problem. Furthermore, the qualitative comparisons are constructive. The paper is generally well-structured, the notations are mostly well-defined, and the experiment results are clearly presented. The proposed algorithm is general and is compatible with existing certification methods.

**Weaknesses:**

- I find the motivation of this work to be a little ambiguous. Specifically, the author argues that the challenge this work aims to address here is "investigating the entire set $\mathcal{F}$ is always a valid but expensive option considering the size of $\mathcal{F}$". However, Algorithm 1 needs to iteratively query the verifier on subsets of $\mathcal{F}$, with the subset in the first couple of iterations potentially quite large. Particularly, at the first iteration, the cardinality of $F_{S_0} \cup F_{S_1}$ seems to be half of $\mathcal{F}$. Therefore, it is not immediately clear to me how the proposed algorithm reduces the overall computational complexity, and some additional explanation would be preferred.
- While the visualizations in Figure 3 are nice, it would be helpful and more meaningful if they could be generated from commonly used model architectures, as ConvNets and vision transformers may attend to different parts of the input image compared with the feed-forward networks used in this paper. Are there any challenges for applying the proposed methods to these more sophisticated networks, since these models usually also have a linear layer as the final layer? I believe that this paper will benefit from a more diverse empirical evaluation.
- Some typos: "uppper" on page 5; "both the baselines" on page 8.

**Questions:**

- Is the certificate issued by ProFIt related to the robust accuracy in any way? My impression after reading the paper is that since $\it{F}_{S_0}$ returned by Algorithm 1 is always sufficient, the percentage of data that can be certified will be the same as the verifier $\mathcal{V}$. Is this correct?
- I see that the experiments and calculations are performed using a CPU. Is the proposed algorithm also compatible with GPU acceleration?

---

> ### Author Response · Authors · 2023-11-16
>
> **Q1. It is not immediately clear to me how the proposed algorithm reduces the overall computational complexity, and some additional explanation would be preferred.**
>
> **R1:** First, let us clarify the statement: "Investigating the entire set $\mathcal{F}$ is always a valid but expensive option, considering the size of $\mathcal{F}$." For DNNs, the proof feature set $\mathcal{F}$ may comprise thousands of proof features, making it impractical to manually investigate each one (e.g., by examining the gradient maps) individually. For this reason, one of our objectives is to automatically extract a proof feature set that is small (the first of three expectations outlined in Section 4.1) and, consequently, easy to manually investigate.
> In Table 1 of the paper, we demonstrate that ProFIt successfully extracts a small yet sufficient and important proof feature set, containing fewer than 10 proof features. This significantly reduces the effort required for the manual investigation of all constituent proof features. We will provide further clarification in the revised version of the paper.
>
> **Q2.  Can ProFIt be used for ConvNets and vision transformers?**
>
> **R2:** All the networks used in the experiments in Table 1 of the paper, are convolutional. We have provided detailed information about the architecture of the DNNs, handled by ProFIt, in the response to Q1 in the common response. For even larger networks like vision transformers, the state-of-the-art deterministic DNN verifiers do not scale. As the scalability of ProFIt is ultimately constrained by the limitations of existing DNN verifiers, it currently does not scale to vision transformers. However, it's crucial to note that ProFIt is a general tool compatible with any deterministic DNN verification algorithm. Therefore, ProFIt will benefit from any future advances to enable the deterministic DNN verifiers to scale to vision transformers. We have discussed this limitation in Section 4.5 of the paper.
>
> **Q3. Is the certificate issued by ProFIt related to the robust accuracy in any way?**
>
> **R3:** ProFIt analyzes the proof only if the underlying verifier $\mathcal{V}$ can prove the property. So, the percentage of cases ProFIt generates proof dissection is the same as the robust accuracy of the DNN with verifier $\mathcal{V}$.
>
> **Q4. Is the proposed algorithm also compatible with GPU acceleration?**
>
> **R4:** Yes, ProFIt is compatible with GPU acceleration. We provide the runtime analysis of ProFIt with GPUs in the response to Q1 in the common response. We will add these runtimes in the revised version of the paper.
>
> **Q5. Typographical errors in page 5 and page 8.**
>
> **R5:** Thanks for pointing out we will fix this in the revised version of the paper.

---

> ### Comment · Reviewer_zRr5 · 2023-11-19
> **Thank you for the response.**
>
> Thank you for the response, which clarified most of my questions. Overall, I think this paper presents a coherent story but the motivation is less clear and prominent. Therefore, I stand on the fence about this manuscript and keep my rating of 6.

---

> > ### Author Response · Authors · 2023-11-21
> >
> > Thank you for your response. Please refer to our reply to reviewer rPr5, particularly addressing the motivation and applications of ProFIt in question 1. For instance, applying ProFIt to DNNs trained with the most recent state-of-the-art robust training method - SABR [1], we observed that while gradient maps for CIFAR10 networks are semantically meaningful, those for MNIST networks are not human-understandable, often highlighting spurious input features from the background (see Appendix C.11). In this case, for MNIST dataset, the SABR training method seems to be overregularizing to achieve higher robustness. For MNIST networks, ProFIt can provide important information about how to pick a hyperparameter value (e.g. value of $\lambda$ in Appendix C of [1]) for training that can avoid overregularization for robustness (as discussed in response to Q2 in common response). The SABR DNNs used in the experiments are taken from the official repository [2].
> >
> > In summary, ProFIt serves as a valuable tool for qualitatively comparing robustness proofs on DNNs trained with different methods, shedding light on the impact of these methods on DNN robustness. Notably, many robust training methods (PGD training, COLT, CROWN-IBP, etc.) prioritize robustness at the expense of standard accuracy [3], making it crucial to analyze why verifiers prove specific properties on a given network.
> >
> >
> > [1] “Certified Training: Small Boxes are All You Need”, M. Mueller, ICLR, 2023. \
> > [2] https://github.com/eth-sri/sabr \
> > [3] "Robustness May Be at Odds with Accuracy", D Tsipras, ICLR 2019.

---

### Official Review · Reviewer_rPr5 · 2023-11-04

**Soundness:** 3 good
**Presentation:** 3 good
**Contribution:** 2 fair
**Rating:** 6
**Confidence:** 3

**Summary:**

This paper aims to analyze the robustness proof generated by a DNN verifier. Given a neural network and a DNN verifier (deterministic, sound, and incomplete verifiers, such as DeepZ and CROWN), the paper presents an algorithm ProFlt to extract a set of proof features (defined as the reachable interval of neurons in the penultimate layer) that is small, sufficient, and retains important proof features. Based on the top-proof feature extracted using the proposed algorithm, the paper then visualizes the gradient maps for different neural networks and DNN verifiers and explains why they differ.

**Strengths:**

The paper is well-structured and technically solid. To the best of my knowledge, most existing works focus on debugging neural networks and explaining how they make predictions. In contrast, this paper proposes to identify the most influential proof features for a DNN verifier, which is different and new. I also like how the paper presents the problem formulation for proof dissection, where the three expectations are discussed in great detail. The proposed ProFit algorithm and how it approximates the priority of proof features are well-explained.

**Weaknesses:**

While I most enjoyed reading the paper, the significance of generating human-understandable proof features and the user case of the extracted proof features need to be explained more clearly. A DNN verifier is supposed to provide a mathematically sound robustness certificate, so by design, it gives a result that can be trusted. However, this paper aims to identify important/human-understandable proof features that can explain how the DNN verifier works, so I wonder why we need a DNN verifier to be explainable in the first place. The following question is: after we obtain the top proof feature (e.g., the one returned by your algorithm), how can we use the extracted insight to assist the DNN verification procedure or the development of adversarially robust models?

Within the proposed problem formulation, the paper did quite a good job explaining the relevant concepts of proof features and developing a well-motivated algorithm to prioritize the proof features. I hope that the authors can better explain the underlying motivation of the considered problem.

**Questions:**

1. What is the computational complexity of the proposed _ProFit_ algorithm and the reliance on the dependent factors?

2. The gradient map visualizations shown in Figure 3 seem interesting. Still, I do not understand the corresponding explanations: "CROWN-IBP filters out most of the spurious features, but it also misses out on some meaningful features." What do you mean by "spurious" and "meaningful" features in the context of gradient maps? Does the provably robust training method CROWN-IBP have a much lower standard accuracy because it overly filters the meaningful features?

3. How can the extracted top proof features benefit the development of better DNN verifiers or robust training methods?


============== Post Rebuttal Comments ==============

I appreciate the authors' feedback and additional experimental results. I would be interested to see how ProFIt analysis can contribute to building more robust DNNs.

---

> ### Author Response · Authors · 2023-11-16
>
> **Q1. Since the DNN verifier is supposed to provide a mathematically sound robustness certificate, explain the motivation for dissecting DNN proofs.**
>
> **R1:** The primary objective of ProFIt is to explain why the verifier successfully proves a property on the given DNN. This motivation aligns with traditional DNN interpretation techniques, where, given a DNN correctly computes the output, the goal is to comprehend the decision-making process of the DNN. In contrast to existing DNN verifiers that often act as black boxes, traditional program verifiers do offer insights into why certain properties are proven for the given program.
> For instance, in programs featuring loops, loop invariants characterize the expected behavior of the program after each iteration, making the proof human understandable (discussed in Section 1 of the paper). We want to check whether the proofs generated by DNN verifiers provide similar insights or not. As shown in Section 5.3, even if a DNN is robust for some $L_{\infty}$ region, the input features, the proof relies on can be different. Moreover, dissection results obtained by ProFIt can serve as a qualitative metric to compare different DNNs achieving similar levels of robustness. For example, proofs generated on standard DNNs highlight spurious input features that are part of the background rather than those associated with the actual object. In contrast, the proof generated on COLT networks relies on input pixels that are part of the actual object.  These applications emphasize the importance of delving into proof dissection techniques.
>
> **Q2.  What is the computational complexity of the proposed  ProFit algorithm and the reliance on the dependent factors?**
>
> **R2:** Please refer to response to Q1 in the common response.
>
> **Q3. What do you mean by "spurious" and "meaningful" features in the context of gradient maps? Does the provably robust training method CROWN-IBP have a much lower standard accuracy because it overly filters the meaningful features?**
>
> **R3:** As defined in Section 5.3 in the paper, for gradient maps generated on MNIST and CIFAR10 images, spurious input features are the pixels coming from the image background, whereas meaningful input features are the pixels that are a part of the actual object being identified by the DNN.
> Robustly trained networks generally report lower standard accuracy than commonly trained networks [1]. Additionally, certifiably robust DNNs (CROWN-IBP) generally report smaller standard accuracy when compared to empirically robust DNNs (PGD-trained) [2]. It appears that the loss in standard accuracy is due to the fact that the certifiably robust networks overly filters the meaningful input features.
>
> [1] "Robustness May Be at Odds with Accuracy", D. Tsipras. ICLR, 2019.
>
> [2] "Certified Training: Small Boxes are All You Need ", M. Mueller, ICLR, 2023.
>
> **Q4 How can the extracted top proof features benefit the development of better DNN verifiers or robust training methods?**
>
> **R4:** Please refer to the response to Q2 in the common response.

---

> > ### Author Response · Authors · 2023-11-21
> >
> > Dear reviewer rPr5,
> >
> > As the discussion period is coming to a close, we would like to take the opportunity to thank you for your time once again. We hope our comments so far have addressed all prior concerns/questions, but please let us know if there is anything we can elaborate on. If you have any other experiments or questions, please let us know and we will do our best to run/address anything before the deadline.

---

### Author Response · Authors · 2023-11-16

Dear Area Chair and Reviewers,


We appreciate the constructive feedback from the reviewers and are encouraged by their acknowledgment of the paper's well-structured presentation, theoretically sound contributions, and detailed experimental validation. We will address and clarify the points raised by the reviewers and will update our paper to reflect this discussion.

First, we will clarify the common concerns raised by multiple reviewers and then address the individual questions.


## Common Response

**Q1. Worst case time complexity analysis and scalability of ProFIt.**

**R1:** In Section 4.3 of the paper, we have already provided a high-level explanation about why the while loop in the ProFIt algorithm always terminates within $O(\log{d\_{l−1})}$ incremental verifier calls. Now, we provide a formal worst-case analysis of the ProFIt and show that our theoretical analysis aligns with our experimental observation in Tables 1 and 2 below.

**Theorem 1: If the size of the penultimate layer is $d\_{l-1}$ then the number of incremental verifier calls made by ProFIt is always $\leq \lceil\log{d\_{l-1}}\rceil + 1$**.


**Proof:** Every iteration of the while loop (Algorithm 1 on page 6 of the paper) makes a single incremental verifier call. So, first, we calculate the maximum number of incremental verifier calls made in the while loop. At the end of each iteration $\boldsymbol{F}\_{S}$ is replaced by either $\boldsymbol{F}\_{S\_1}$ (line 14 of Algorithm 1) or $\boldsymbol{F}\_{S\_2}$ (line 17 of Algorithm 1). Now, $\max(|\boldsymbol{F}\_{S\_1}|, |\boldsymbol{F}\_{S\_2}|) \leq (|\boldsymbol{F}\_{S}| + 1) /2$ where $|\cdot|$ denotes the cardinality of a set. Initially, $\boldsymbol{F}\_{S} = \mathcal{F}$ and $|\boldsymbol{F}\_{S}| = |\mathcal{F}| = d\_{l-1}$. Then after $i$th iteration of the while loop the following inequality holds $|\boldsymbol{F}\_{S}| \leq \frac{d\_{l-1}}{2^i} + \sum\_{j=1}^{i} \frac{1}{2^j}$. Let, $I$ denote the total number of iterations of the while loop. Given the while loop only executes when $\boldsymbol{F}\_{S}$ is non-empty (line 9 of Algorithm 1), $1 \leq \frac{d\_{l-1}}{2^{(I-1)}} + \sum\_{j=1}^{I-1} \frac{1}{2^{j}} \implies (I -1) \leq \lceil\log{d\_{l-1}}\rceil \implies I \leq \lceil\log{d\_{l-1}}\rceil + 1$.


Hence, the total number of incremental verifier calls is always $\leq \lceil\log{d_{l-1}}\rceil + 1$. Note, since we only modify the final layer we run incremental verification that avoids rerunning verification on the entire network from scratch. However, computing proof features (line 6 of Algorithm 1) requires a single verifier call which can not be run incrementally.

For a network with $n$ neurons and $l$ layers a single incremental DeepZ [9] verifier call in the worst case takes $O(n^3)$ time whereas a single call to incremental CROWN [10] verifier in the worst case takes $O(l \times n^3)$ time. The cost for a single non-incremental DeepZ verifier call is in the worst case $O(l \times n^3)$ whereas for CROWN it is $O(l^2 \times n^3)$. Overall, the complexity of ProFIt with CROWN is $O((\log{d_{l-1}} + l) \times l \times n^{3})$. We will update our paper with detailed proof of the worst-case runtime of ProFIt.

---

> ### Author Response · Authors · 2023-11-16
>
> **Experimental evaluation of ProFIt runtime:**
>
>
>
> - **Experiment setup:** We run ProFIt on multiple networks including those mentioned in the paper on both CPUs and GPUs. For CPU-related experiments, we use the same setup as mentioned in Section 5.1 in the paper. In GPU-related experiments, we utilize a single NVIDIA A100-PCI GPU with 40 GB RAM. We use the state-of-the-art incomplete verifier $\alpha$-CROWN from auto_LiRPA [6] toolbox and show results for 100 local $L_{\infty}$ robustness properties. As done in the paper, for MNIST and CIFAR10 networks, we use $\epsilon = 0.02$ and $\epsilon =0.2/255$ respectively to define $L_{\infty}$ input regions.
>
>
> - **Size and architecture of the Networks:** We present the runtime analysis of ProFIt on the DNNs used in the paper and also showcase results on some of the largest verifiable network architectures, namely ConvBig and ConvSuper. These architectures are de facto benchmarks employed for testing the scalability of state-of-the-art verifiers [1, 2, 3] and are featured in the International Verification of Neural Networks Competition [4]. All the networks used in this experiment are selected from the ERAN [8], and CROWN-IBP [11] repositories and belong to the category of convolutional neural networks (CNNs).
>
>
>
>
>
>
> - **Compatibility with GPU acceleration:** ProFIt is not restricted to CPUs and works with DNN verifiers whose computation can benefit from GPU acceleration. For example, ProFIt can be run with GPU implementation of auto_LiRPA [6]. We show the runtime improvement of ProFIt with GPUs below. Runtimes on GPUs are up to 20x shorter than those on CPUs.
>
> - **Experimental Results:** We present results for MNIST networks in Table 1 and results for CIFAR networks in Table 2. Column 1 displays the network name, column 2 outlines the training method, and columns 3 and 4 describe the network's structure. Column 5 indicates the worst-case bound on the number of incremental verifier calls, as described in Theorem 1, while column 6 displays the average number of incremental verifier calls per property. Columns 7 and 8 present the average runtime in seconds for ProFIt on CPU and GPU, respectively. Notably, on GPUs, the average runtime for ProFIt is less than 1 second for all the networks. Note that, for ConvBig architectures, the timeout used in the International Verification of Neural Networks Competition [4] is in minutes. In comparison, ProFIt with $\alpha$-CROWN is significantly faster.
>
>  #### Table 1 on MNIST Networks
> | Network | Training Method |# Layers |# Params | Theoretical bound|Avg. verifier calls |Avg. CPU time | Avg. GPU time |
> |:---------|:---------|:---------|:---------|:---------|:---------|:---------|:---------|
> |ConvSmall| Standard|4|90K|9|7.81|0.85s|0.14s|
> |ConvMed| PGD|5|200K|12|9.90|1.80s|0.16s|
> |ConvSmall| COLT|4|90K|9|7.25|0.39s|0.11s|
> |IBP-Small| CROWN-IBP|4|80K|9|6.99|0.34s|0.09s|
> |ConvBig|DiffAI [7]|7|1.8M|11|7.12|2.57s|0.29s|
> |ConvSuper|DiffAI [7]|7|10M|11|7.08|6.28s|0.30s|
>
>
>  #### Table 2 on CIFAR10 Networks
> | Network | Training Method | # Layers | # Params | Theoretical bound|Avg. verifier calls |Avg. CPU time | Avg. GPU time |
> |:---------|:---------|:---------|:---------|:---------|:---------|:---------|:---------|
> |ConvSmall| Standard|4|120K|9|5.36|0.30s|0.07s|
> |ConvSmall| PGD|4|120K|9|6.04|0.32s|0.09s|
> |ConvSmall| COLT|4|120K|10|5.64|0.52s|0.15s|
> |IBP-Small| CROWN-IBP|4|100K|10|6.2|0.48s|0.11s|
> |ConvBig|DiffAI [7]|7|2.5M|11|7.8|3.93s|0.50s|
>
>
>
>
> [1] “PRIMA: General and Precise Neural Network Certification via Scalable Convex Hull Approximations”, M. Muller, POPL, 2022.
>
> [2] “Beta-CROWN: Efficient Bound Propagation with Per-neuron Split Constraints for Neural Network Robustness Verification”, C. Ferrari, NeurIPS 2021.
>
> [3] "Multi-neuron relaxation guided branch-and-bound", C. Wang, ICLR 2022.
>
> [4] [https://sites.google.com/view/vnn2023](https://sites.google.com/view/vnn2023)
>
> [5] "Scaling Polyhedral Neural Network Verification on GPUs", C. Muller, MLSys, 2021.
>
> [6] https://github.com/Verified-Intelligence/auto_LiRPA
>
> [7] "Differentiable Abstract Interpretation for Provably Robust Neural Networks", M. Mirman, ICML, 2018.
>
> [8] https://github.com/eth-sri/eran
>
> [9] "Fast and Effective Robustness Certification", G. Singh, NeurIPS 2018.
>
> [10] "Efficient Neural Network Robustness Certification with General Activation Functions", H. Zhang, NeurIPS 2018.
>
>  [11] https://github.com/huanzhang12/CROWN-IBP

---

> > ### Author Response · Authors · 2023-11-16
> >
> > **Q2. Applying ProFIt analysis for training robust DNNs:**
> >
> >
> >
> > **R2:** The ProFIt analysis can be leveraged in the following ways to train robust DNNs.
> >
> > **Debugging and avoiding overregularization:** As shown in Section 5.3, ProFIt analysis can be used to detect whether the model is overregularizing to achieve higher robustness. Many certifiable training methods (COLT, CROWN-IBP, etc.) rely on hyperparameters to strike a balance between standard accuracy and robustness. ProFIt serves as a valuable tool for comprehending the impact of different hyperparameters in the tradeoff between standard accuracy and robustness [1] of trained DNNs. In Appendix C.10, we utilize ProFIt to analyze the influence of the hyperparameter $\epsilon_{train}$ - used to define the $L_{\infty}$ region during training, on the robustness proof generated on the trained DNN. As $\epsilon_{train}$ increase, we observe that the proof generated on the DNNs trained with COLT and PGD training filter out more input features.
> >
> >
> >
> > **Priority of neurons:** ProFIt assigns priority to each neuron in the penultimate layer, and in Appendix C.3, we demonstrate that lower-priority neurons typically capture spurious input features. ProFIt can be employed during training to ensure that lower-priority neurons do not get activated. However, this needs further investigation into how to incorporate ProFIt during training time.
> >
> > **Data augmentation:** The generated visualization i.e. gradient map of the top proof features can directly be used to train better models through data augmentation [2, 3].
> >
> > [1] "Robustness May Be at Odds with Accuracy", D Tsipras, ICLR 2019.
> >
> > [2] "SaliencyMix: A Saliency Guided Data Augmentation Strategy for Better Regularization", A . Uddin, ICLR, 2021.
> >
> > [3] "Saliency Map-Based Data Augmentation", Al-Afandi et. al. ICPR, 2022.

---

> > > ### Author Response · Authors · 2023-11-21
> > >
> > > Dear Area Chair and Reviewers,
> > >
> > > We would like to thank the reviewers once again for their constructive feedback.  We have updated the PDF with the changes suggested by the reviewers including a formal worst-case runtime analysis of ProFIt (Appendix D) and experiments showcasing the scalability of the proposed framework (Appendix E). Section 4.5 has been revised to explain ProFIt's limitations with probabilistic verifiers and very large networks like vision transformers. If there are any other experiments or questions you have please let us know and we will do our best to run/address anything before the deadline.

---

### Meta-Review · Area_Chair_aT5L · 2023-12-08

**Metareview:**

The authors develop a technique for interpreting formal verification proofs of the robustness of neural networks to adversarial examples, and provide an algorithm that computes, for any verification algorithm, any minimal set of "features" (neurons at the penutimate layer) that are crucial to analyze in the robustness proof. They run their approach on image classification networks trained to be robust, either in a certifiable or in an empirical sense and find interesting patterns in the proofs that emerge. While the immediate utility of this analysis is unclear, reviewers agree that this could make an interesting contribution to the literature and spark further research on the nature of neural network verifiers. Hence I recommend acceptance.

**Justification For Why Not Higher Score:**

The utility of interpreting neural network robustness proofs is unclear, and the paper does not provide sufficient motivation for the same.

**Justification For Why Not Lower Score:**

The paper brings up an interesting perspective to analyze neural network robustness proofs that could spark further ideas in analyzing and developing stronger neural network verifiers. Hence, I think the paper would be worth presenting to the ICLR audience.

---

### Decision · Program_Chairs · 2024-01-16

Accept (poster)